# Development of optimized drug-like small molecule inhibitors of the SARS-CoV-2 3CL protease for treatment of COVID-19

Hengrui Liu [1], Sho Iketani [2,3], Arie Zask[4], Nisha Khanizeman[1], Eva Bednarova[1], Farhad Forouhar[5], Brandon Fowler[1], Seo Jung Hong[6], Hiroshi Mohri[2], Manoj S. Nair [2], Yaoxing Huang [2], Nicholas E. S. Tay[1], Sumin Lee[1], Charles Karan[7], Samuel J. Resnick[6,8], Colette Quinn[9], Wenjing Li[9], Henry Shion[9], Xin Xia[4], Jacob D. Daniels[10], Michelle Bartolo-Cruz[4], Marcelo Farina[4,11], Presha Rajbhandari[4], Christopher Jurtschenko[9], Matthew A. Lauber[9], Thomas McDonald[9], Michael E. Stokes[4], Brett L. Hurst [12], Tomislav Rovis [1✉], Alejandro Chavez [6✉], David D. Ho [2✉] & Brent R. Stockwell [1,4✉]

The SARS-CoV-2 3CL protease is a critical drug target for small molecule COVID-19 therapy, given its likely druggability and essentiality in the viral maturation and replication cycle. Based on the conservation of 3CL protease substrate binding pockets across coronaviruses and using screening, we identified four structurally distinct lead compounds that inhibit SARS-CoV-2 3CL protease. After evaluation of their binding specificity, cellular antiviral potency, metabolic stability, and water solubility, we prioritized the GC376 scaffold as being optimal for optimization. We identified multiple drug-like compounds with <10 nM potency for inhibiting SARS-CoV-2 3CL and the ability to block SARS-CoV-2 replication in human cells, obtained co-crystal structures of the 3CL protease in complex with these compounds, and determined that they have pan-coronavirus activity. We selected one compound, termed coronastat, as an optimized lead and characterized it in pharmacokinetic and safety studies in vivo. Coronastat represents a new candidate for a small molecule protease inhibitor for the treatment of SARS-CoV-2 infection for eliminating pandemics involving coronaviruses.

[1] Department of Chemistry, Columbia University, New York, NY 10027, USA. [2] Aaron Diamond AIDS Research Center, Columbia University Irving Medical Center, New York, NY 10032, USA. [3] Department of Microbiology and Immunology, Columbia University Irving Medical Center, New York, NY 10032, USA. [4] Department of Biological Sciences, Columbia University, New York, NY 10027, USA. [5] Herbert Irving Comprehensive Cancer Center, Columbia University Irving Medical Center, New York, NY 10032, USA. [6] Department of Pathology and Cell Biology, Columbia University Irving Medical Center, New York, NY 10032, USA. [7] Sulzberger Columbia Genome Center, Columbia University, New York, NY 10032, USA. [8] Medical Scientist Training Program, Columbia University Irving Medical Center, New York, NY 10032, USA. [9] Waters Corporation, 34 Maple Street, Milford, MA 01757, USA. [10] Department of Pharmacology and Molecular Therapeutics, Columbia University Irving Medical Center, New York, NY 10032, USA. [11] Department of Biochemistry, Federal University of Santa Catarina, Florianópolis, Santa Catarina, Brazil. [12] Institute for Antiviral Research, Utah State University, Logan, UT 84322, USA. ✉email: tr2504@columbia.edu; ac4304@cumc.columbia.edu; dh2994@cumc.columbia.edu; bstockwell@columbia.edu

S ARS-CoV-2 is the viral pathogen responsible for the global COVID-19 pandemic, which has caused millions of infections and more than five million deaths worldwide to date[1–3]. Although the speed with which COVID-19 vaccines have been developed is remarkable, their long-term protection effect and effectiveness against emerging variants and potential future variants of SARS-CoV-2 and other coronaviruses remains to be determined[4–8]. Considering the limited distribution and effectiveness of current vaccines, direct-acting anti-viral therapeutics are needed. As viral protease inhibitors have been transformative therapies for viral infections such as HIV and HCV[9–11], the essential role of 3CL protease ($M^{pro}$) in the maturation and replication of SARS-CoV-2 makes it an attractive target for COVID-19 therapy[12–18]. Given that the 3CL protease substrate-binding pocket is highly similar across 12 different coronaviruses, we hypothesized that small-molecule inhibitors of coronavirus 3CL proteases may be useful starting scaffolds for the design of inhibitors of the SARS-CoV-2 3CL protease[14]. Moreover, due to the conservation of 3CL protease substrate-binding pockets across coronaviruses, optimized 3CL protease inhibitors have the potential to be broadly effective against current variants and potential future variants of SARS-CoV-2 and other coronaviruses.

We validated this hypothesis and identified multiple small molecules effective against the SARS-CoV-2 3CL protease, but initially with moderate potencies and suboptimal selectivity. We characterized their binding mechanisms to the 3CL protease, which provided the foundation for structure-based optimization of these compounds, leading to the discovery of novel chemical entities that function as potent and selective 3CL protease inhibitors. These optimized compounds featured improved potency and selectivity in comparison with the SARS-CoV-2 3CL protease inhibitors reported by other groups[13–28] and exhibited pan-coronavirus potency; together, these compounds revealed insights into the design of drug-like 3CL protease inhibitors. These studies resulted in creation of a novel small-molecule SARS-CoV-2 3CL protease inhibitor, termed *coronastat*, as a development candidate for treating COVID-19, and provide a roadmap to the rapid discovery of treatments for future pandemic viruses.

## Results

**Identification of lead compounds for the development of SARS-CoV-2 3CL protease inhibitors.** In line with the hypothesis that inhibitors of other viral proteases may be effective against the novel SARS-CoV-2 3CL protease, we found that the SARS-CoV 3CL protease inhibitors known as compound 4 (ref. [29]), GC376 (ref. [30]), MAC5576 (ref. [31]), and the more general cysteine protease inhibitor and anti-oxidant ebselen[32], inhibited the activity of SARS-CoV-2 3CL protease on a specific fluorogenic substrate, with modest $IC_{50}$ values of 123 nM, 101 nM, 93 nM, and 192 nM, respectively (Fig. 1).

In addition to these four lead compounds, aiming to find additional scaffolds that inhibit this viral protease, we screened a library of 7247 diverse electrophile fragments for inhibitory activity against the SARS-CoV-2 3CL cysteine protease. Further evaluation of the resulting 55 hits suggested inhibition potency more than 10-fold poorer than GC376 in the same assay (Supplementary Figs. 1, 2). To improve potency, inspired by the 2-mercaptopyridine and benzene-sulfonamide scaffolds, which frequently appeared in fragment screening hit structures, we synthesized and tested additional cysteine-reactive electrophiles, including 2-sulfonylpyridines[33] and sulfonamide derivatives against the SARS-CoV-2 3CL protease (Supplementary Fig. 3a, b). However, no further improvements in the potency for protease inhibition were observed with modifications of these screening hits. Therefore, we prioritized the initial four lead compounds for further evaluation (Supplementary Fig. 3c, d).

**Evaluation of lead compounds for binding to SARS-CoV-2 3CL protease.** In addition to testing potency in the protease biochemical assay, we evaluated these four candidate compounds for their binding activity to the 3CL protease, their cellular antiviral potencies, and their metabolic stability.

First, using isothermal titration calorimetry (ITC), we determined the binding stoichiometry of compound 4 (0.97 ± 0.06), GC376 (1.2 ± 0.2), and MAC5576 (1.2 ± 0.2) to the SARS-CoV-2 3CL protease (Supplementary Fig. 4). The 1:1 stoichiometry detected indicates specificity for a single binding site. Binding of ebselen, however, didn't reach saturation until a high molar ratio in the ITC assay, which suggested non-specific binding on multiple surface cysteines. A fragment analog of ebselen, AZVIII-40A, with sulfur replacing the more reactive selenium atom to reduce reactivity towards non-specific surface cysteines, was tested, but it exhibited a lower potency than ebselen for inhibition of the SARS-CoV-2 3CL protease, with an $IC_{50}$ value of 790 nM, suggesting the necessity of the more reactive selenium-containing warhead (Supplementary Fig. 5).

We used mass spectrometry to investigate the binding modes of these four scaffolds. We started with the evaluation of the binding mode of compound 4. A one-to-one complex between compound 4 and the SARS-CoV-2 3CL protease was detected by MALDI MS analysis of the intact protein after incubation with compound 4, with a mass shift matching the molecular weight of compound 4 (Supplementary Fig. 6a). This suggests irreversible covalent binding of a single compound 4 to the 3CL protease via a thiol Michael addition. Furthermore, LC-MS analysis of the digested 3CL-compound 4 complex confirmed that compound 4 specifically binds in one-to-one stoichiometry, and that this binding selectively occurs on the active site cysteine, C145, of the SARS-CoV-2 3CL protease (Supplementary Fig. 6b).

We next investigated the binding mode of the compound GC376 to the 3CL protease. MALDI MS and reverse-phase LC-ESI-MS (RPLC-MS) at pH 2 could not detect the covalent binding of GC376, the binding of which is likely reversible and sensitive to pH, based on the masked aldehyde functionality of GC376 (Supplementary Fig. 7a). However, we discovered that pre-incubation of the 3CL protease with one equivalent of GC376 completely blocked the subsequent binding of compound 4 to 3CL, indicating that GC376 binds to the same active site of the SARS-CoV-2 3CL protease (Supplementary Fig. 7b). Additionally, pre-incubation of 3CL with one equivalent of GC376 also blocked covalent modification of 3CL with cysteine-reactive iodoacetamide, as detected by RPLC-MS. Moreover, native size exclusion chromatography ESI-MS (SEC-MS) was used to detect the binding of two GC376 molecules to the dimer of the 3CL protease at pH 7. The mass shift induced by GC376 in the native SEC-MS suggested that the bisulfite moiety on GC376 transforms into an aldehyde warhead before the subsequent formation of a thiohemiacetal upon covalent binding to cysteine 145 on the 3CL protease (Supplementary Fig. 7c). This binding chemistry is consistent with the pH-sensitive reversible covalent binding of GC376 and the observation that GC373 (AZVIII-58), an aldehyde analog of GC376, can also covalently modify cysteine 145 on the 3CL protease, and prevent the binding of iodoacetamide to 3CL (Supplementary Fig. 7b).

We then evaluated the binding mode of ebselen; as noted above, we had determined in the ITC assay that ebselen does not bind in a one-to-one stoichiometry. Indeed, we detected the binding of multiple ebselen molecules to each 3CL protease molecule by RPLC-MS and native SEC-MS, which was consistent with ITC results demonstrating non-specific binding (Supplementary Fig. 7c). This finding reinforced the conclusion that ebselen would not be a promising starting scaffold for a

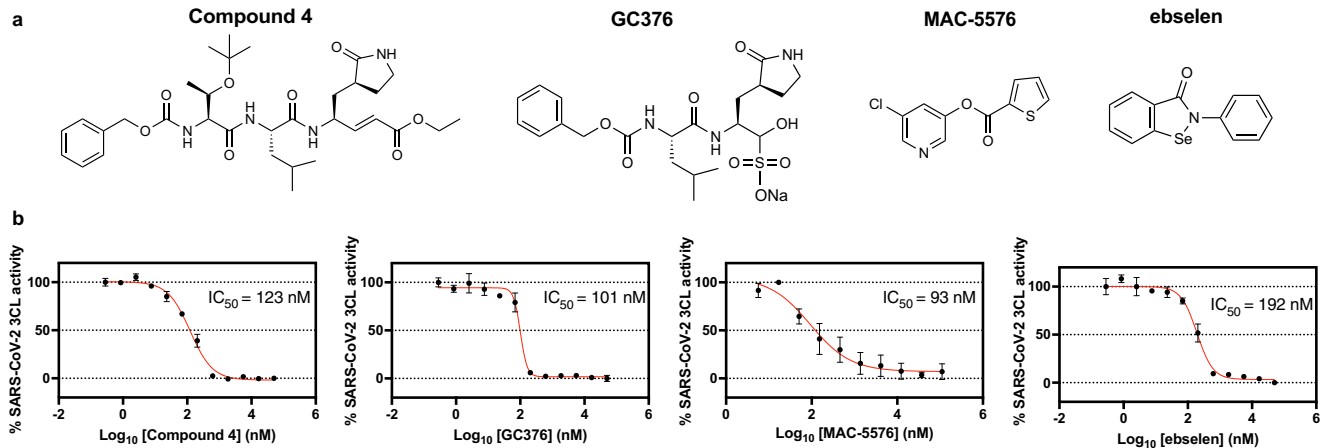

**Fig. 1 Compound 4, GC376, MAC-5576, and ebselen inhibited SARS-CoV-2 3CL protease. a** Structures of compound 4, GC376, MAC-5576, and ebselen. **b** The dose-dependent effects of compound 4, GC376, MAC-5576, and ebselen on the activity of 200 nM SARS-CoV-2 3CL protease were tested. The fluorogenic peptide MCA-AVLQSGFR-Lys(DNP)-Lys-NH2, corresponding to the nsp4/nsp5 cleavage site in the virus was applied as substrate. Data are plotted as the mean ± s.d., $n$ = 3 biological replicates. Source data are provided as a Source Data file.

therapeutically useful SARS-CoV-2 3CL protease inhibitor, despite prior reports on its activity.

Finally, we evaluated the binding mode of MAC-5576: we detected a one-to-one complex with SARS-CoV-2 3CL protease by MALDI MS, with a mass shift matching the carbonyl-thiophene moiety of MAC-5576, which suggested covalent binding of a single MAC-5576-derived moiety to the 3CL protease via thiol-transesterification (Supplementary Fig. 8a). In addition, LC-MS analysis of the digested 3CL-MAC-5576 complex revealed that MAC-5576 selectively binds to the active site cysteine 145 of the SARS-CoV-2 3CL protease. Despite its low molecular weight, MAC-5576 is a selective and specific inhibitor of the SARS-CoV-2 3CL protease (Supplementary Fig. 8b).

We next examined the binding selectivity of the four candidates for the viral protease over human proteases to evaluate potential off-target effects that could lead to undesirable toxicities. Considering that no human protease has been found to share similar cleavage specificity as SARS-CoV-2 3CL protease, identification of homologous human proteases has been challenging, although the difference in structure makes 3CL protease an attractive therapeutic target[13]. From a structural similarity perspective, we identified human chymotrypsin as the closest structural homolog of SARS-CoV-2 3CL protease irrespective of the catalytic mechanism, as it shares the same two β-barrel fold as 3CL protease[31]. Accordingly, we tested the effects of the four candidates on human chymotrypsin, none of which showed inhibitory effects on chymotrypsin, suggesting high selectivity for the viral protease (Supplementary Fig. 9).

In summary, while all four compounds selectively bound to the SARS-CoV-2 3CL protease and inhibited protease activity, ebselen was not assigned priority for further development due to its low specificity.

**Evaluation of lead compounds for cellular antiviral potency.** Upon testing these compounds for inhibition of SARS-CoV-2 viral replication in Vero-E6 cells, we found that compound 4, GC376, and ebselen blocked viral infection, with $EC_{50}$ values of 0.8 μM, 6 μM, and 0.8 μM, respectively, whereas MAC-5576 did not (Fig. 2a).

With a goal of improving the cellular potency of MAC-5576, we further tested a series of its analogs (Supplementary Fig. 10a, b). Since the $IC_{50}$ value of MAC-5576 (93 nM) already was at the detection limit of our initial biochemical assay with 200 nM 3CL

protease, for analogs showing comparable biochemical-$IC_{50}$ values, we further measured their $IC_{50}$ value using 20 nM 3CL protease, followed by a comparison with MAC-5576 side by side ($IC_{50}$ = 10 nM, Supplementary Fig. 10c). We identified three analogs of MAC-5576 capable of inhibiting the SARS-CoV-2 3CL protease activity in vitro: T71, Z43, and Z91, with $IC_{50}$ values of 11 nM, 14 nM, and 31 nM, respectively. Using ITC, we confirmed Z43 binds to the SARS-CoV-2 3CL protease with 1:1 stoichiometry, indicating specificity for a single binding site (Supplementary Fig. 10d). Considering all MAC-5576 analogs tested, we concluded that bromine can replace chlorine in the recognition element (leaving group) and five-membered heterocycles were preferred in the acyl group, where a thiazole worked better than thiophene or furan. Although having favorable in vitro potencies, these MAC-5576 analogs were still not active in a mammalian cell-based assay[34] for inhibition of the SARS-CoV-2 3CL protease, potentially due to stability or cell permeability issues (Supplementary Fig. 10e). Therefore, despite its specific binding and potent biochemical inhibition against the SARS-CoV-2 protease enzyme activity, MAC-5576 was not a promising candidate for development due to its low cellular activity.

**Evaluation of lead compound metabolic stabilities.** The preceding evaluation suggested that MAC-5576 and ebselen would not be suitable starting points for developing drug-like inhibitors of the SARS-CoV-2 3CL protease, despite their biochemical activity against the protease enzyme activity. Meanwhile, the data suggested the suitability of compound 4 and GC376 as possible lead compounds for the further development of SARS-CoV-2 3CL protease inhibitors.

Since compound 4 and GC376 exhibited moderate potencies, but high selectivity and specificity, we further tested their metabolic stabilities, which are often predictive of pharmacokinetics in animals and humans[35,36]. Compound 4 and GC376 were both stable in human plasma ($T_{1/2}$ > 240 min). GC376 was also stable in human liver microsomes ($T_{1/2}$ > 80 min), which reflects liver metabolism, whereas compound 4 had only moderate stability in human microsomes ($T_{1/2}$ = 19 min, Fig. 2b and Supplementary Fig. 11a, b). GC376 was also stable in mouse plasma and microsomes ($T_{1/2,plasma}$ > 240 min, $T_{1/2,microsome}$ > 80, Supplementary Fig. 11c, d). In contrast, compound 4 exhibited low mouse plasma and microsomal stability independent of NADPH, suggesting the involvement of enzymes other than

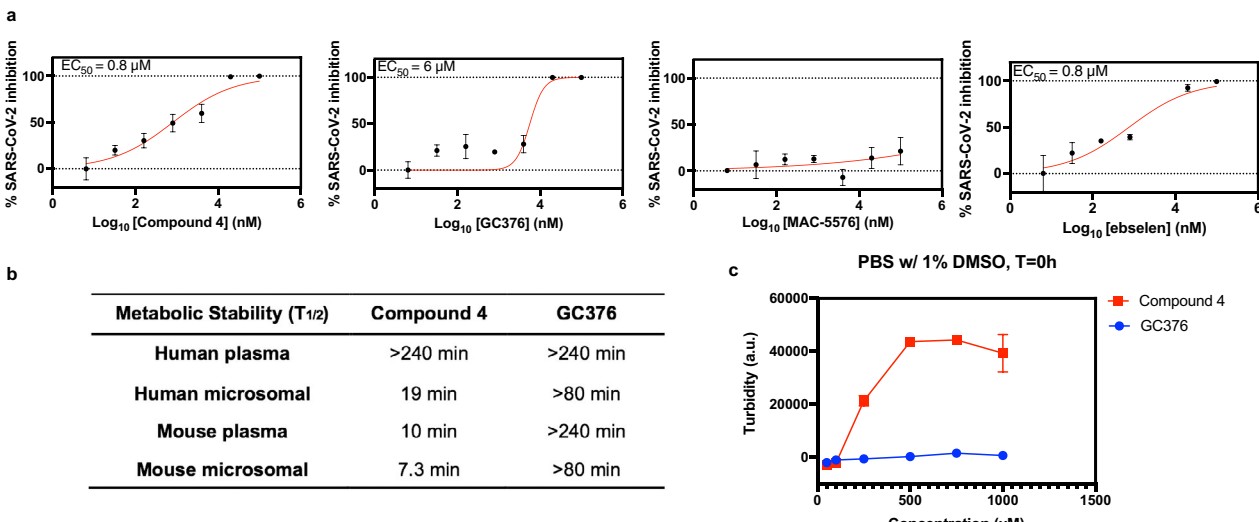

**Fig. 2 Evaluation of lead compounds for cellular antiviral potency, metabolic stability, and solubility. a** Ability of compound 4, GC376, MAC-5576, and ebselen to inhibit SARS-CoV-2 viral infection in cell culture. Stocks of SARS-CoV-2 strain 2019-nCoV/USA_WA1/2020 were propagated and titered in Vero-E6 cells. Serial dilutions of the test compound were prepared in cell media (EMEM + 10% FCS + penicillin/streptomycin), overlaid onto cells, and then virus was added to each well at an MOI (multiplicity of infection) of 0.2. Cells were incubated at 37 °C under 5% $CO_2$ for 72 h before viral RNA extraction from the supernatant and quantification against a RNA standard by quantitative reverse-transcriptase PCR (qRT-PCR). Data are plotted as the mean ± s.d., $n = 3$ biological replicates. **b** Half-life of compound 4 and GC376 in human and mouse (CD-1) plasma and liver microsomes (with NADPH). **c** Solubility test (light scattering) of compound 4 and GC376 in PBS with 1% DMSO. Data are plotted as the mean ± s.d., $n = 3$ biological replicates. Source data are provided as a Source Data file.

cytochrome P450s (CYPs) driving compound 4 metabolic conversion ($T_{1/2,plasma} = 10$ min, $T_{1/2,microsome} = 7.3$ min, Supplementary Fig. 11e, f). Moreover, solubility tests demonstrated that compound 4 formed precipitates above 100 μM, while GC376 was soluble in 1% DMSO PBS buffer at up to 1 mM concentrations (Fig. 2c). Although turbidity was observed in aqueous solutions of 25–200 mM GC376, the solution became clear at 400 mM and above (Supplementary Fig. 12), potentially due to micelle formation as reported in a recent study of GC376[24]. This suggested that GC376 featured promising drug-like properties and that compound 4 would be a challenging scaffold to develop into a therapeutic agent.

To further understand the route of metabolism of compound 4, we identified metabolites of compound 4 using LC-MS/MS analysis (Supplementary Fig. 13a). The ester-hydrolyzed product was observed in both mouse and human microsome samples and its formation may not be CYP-mediated, as it is also found in mouse plasma samples (Supplementary Fig. 13b). The carboxybenzyl-cleaved metabolite was observed in human and mouse microsome samples (Supplementary Fig. 13c). While there were no fragment ions directly pointing to an intermediate oxidation of the benzyl group, it is suspected, given the similarity of the MS/MS spectra in Supplementary Fig. 13c, d, despite their different parent ions. Furthermore, the carboxybenzyl-cleaved metabolite was only formed in microsome samples in the presence of NADPH, suggesting that CYPs are involved in the generation of this metabolite. Additional oxidized products of compound 4 were also observed: a γ-lactam-oxidized product and a leucine-oxidized product, both of which were observed in mouse and human microsome samples (Supplementary Fig. 13e, f). This analysis provided insights for further optimization of peptidic 3CL protease inhibitors.

To optimize the structure of compound 4 to overcome its metabolic limitation, we first found that the chirality of the carbon bearing the lactam moiety was critical, as AZVIII-34D, with an (R)-configuration at this carbon instead of the (S)-configuration in compound 4 exhibited a poorer $IC_{50}$ value of 9.7 μM

(Supplementary Fig. 14a–c). Considering the structural similarities of compound 4 and GC376, we tried to transplant the Michael-receptor warhead of compound 4 onto the GC376 scaffold, but the resulting compound, termed AZVIII-41A, was much less active, with an $IC_{50}$ value > 50 μM (Supplementary Fig. 14d). As compound 4 is bulkier than GC376 and exhibited lower metabolic stability, we attempted to simplify the structure via removal of the N-terminal amino acid group, but the abbreviated analog was surprisingly inactive (AZVIII-33B), as was the corresponding alcohol (AZVIII-37A). Additional removal of the leucine residue combined with substitutions of the warhead with an ebselen-like warhead (AZVIII-43A), ester (AZVIII-38), or alcohol (AZVIII-30) did not produce active compounds, while the aldehyde warhead functioned slightly better in this series of compounds (AZVIII-42). Therefore, no analog of compound 4 with better or comparable potency was identified to compensate for its limited metabolic stability.

In summary, despite its promising potency against the SARS-CoV-2 3CL protease, compound 4 was not suitable for development of a COVID-19 development candidate due to its low metabolic stability and difficulty in further optimizing this scaffold.

**Structural optimization of the GC376 led to discovery of analogs with high potency for inhibiting SARS-CoV-2.** We then focused on the creation of improved GC376 analogs, due to its specific binding, high antiviral potency, and metabolic stability. To optimize the structure of GC376, we used a structure-based approach. Schrodinger Suite/Desmond molecular dynamics calculations generated a low energy solution conformation of GC376, the peptide backbone and lactam ring of which closely aligned with a bound x-ray structure of GC376 that we solved[37] (Supplementary Fig. 15). This showed not only a high degree of pre-organization of the compound for binding, which is consistent with a favorable entropy of binding, but also flexibility of benzyl and isopropyl groups. The relatively weak electron densities and lack of strong intermolecular interactions of the benzyl and isopropyl groups in the co-crystal sctructure of GC376 with

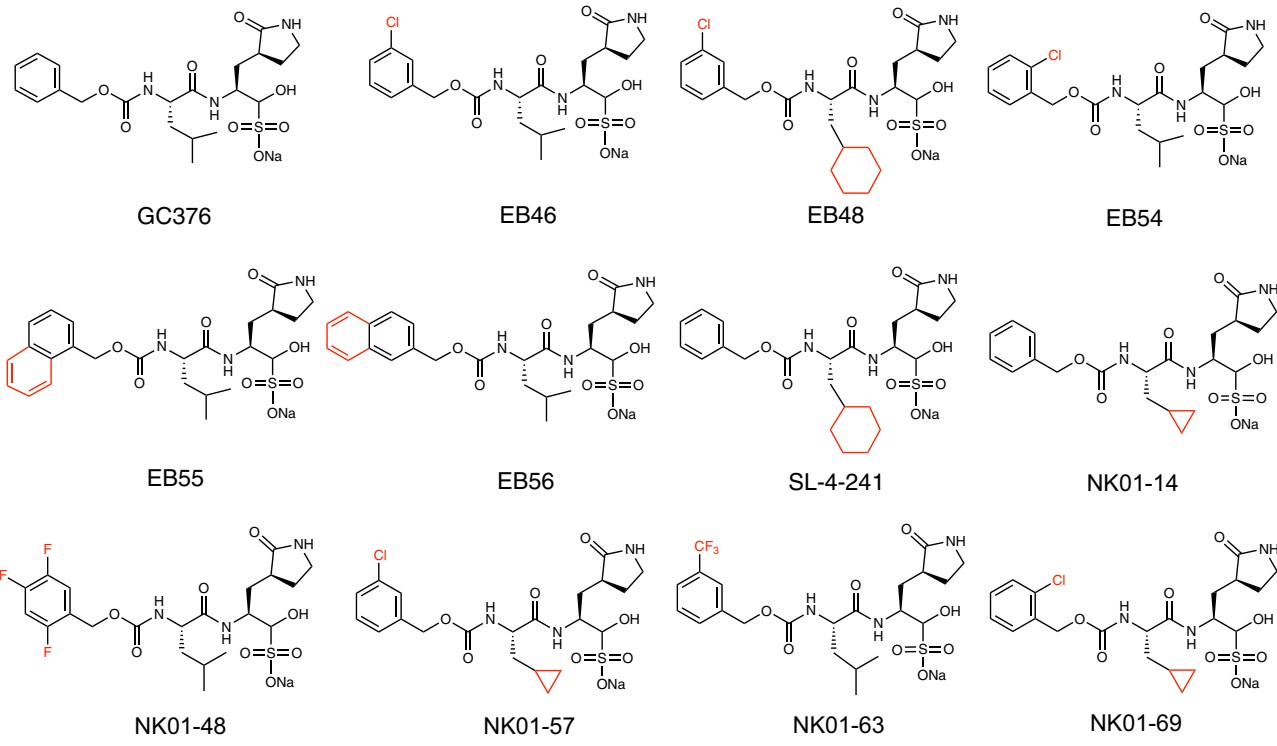

**Fig. 3 Analogs of GC376 developed as SARS-CoV-2 3CL protease inhibitors.** Structures of GC376 and its analogs were shown, with modifications highlighted in red.

**Table 1 SARS-CoV-2 inhibition data for GC376 analogs.**

| Entry | SARS-CoV-2 3CL$^{pro}$ IC$_{50}$ (nM) | SARS-CoV-2 3CL$^{pro}$ $k_{inact}/K_I$ (min$^{-1}$ μM$^{-1}$) | SARS-CoV-2 Vero E6 EC$_{50}$ (μM) | SARS-CoV-2 Huh-7$^{ACE2}$ EC$_{50}$ (nM) | SARS-CoV-2 Caco-2 EC$_{90}$ (nM) |
|---|---|---|---|---|---|
| GC376 | 35 | 1.3 | 9 | 51 | |
| EB46 | 23 | 3.3 | 2 | 14 | 53 |
| EB48 | 200 | 2.1 | 16 | | |
| EB54 | 29 | 5.3 | 2 | 5 | 190 |
| EB55 | 72 | 1.3 | 3 | 11 | |
| EB56 | 50 | 1.1 | 4 | 28 | |
| SL-4-241 | 75 | 1.4 | 17 | | |
| NK01-14 | 22 | 4.3 | | 26 | |
| NK01-48 | 92 | 0.3 | | 15 | |
| NK01-57 | 13 | NA | | 92 | |
| NK01-63 | 16 | 4.4 | | 6 | 81 |
| NK01-69 | 24 | 4.2 | | 35 | |

the 3CL protease also suggested the feasibility of modifications of these two groups to optimize the structure. Therefore, we designed and synthesized 11 analogs of GC376 with various modifications on the benzyl and isopropyl groups (Fig. 3).

Our evaluation of these analogs started from the measurement of their inhibition potency using 20 nM SARS-CoV-2 3CL protease in the biochemical assay (Supplementary Fig. 16). Compared with GC376 in the same assay (IC$_{50}$ = 35 nM), six analogs featured improved potency: EB46, EB54, NK01-14, NK01-57, NK01-63, and NK01-69, with IC$_{50}$ values of 23 nM, 29 nM, 22 nM, 13 nM, 16 nM, and 24 nM, respectively (Table 1). To compare these analogs with the four initial lead compounds, we evaluated cellular antiviral potency of a subset of analogs in Vero-E6 cells (Supplementary Fig. 17a). Compared with GC376 (EC$_{50}$ = 9 μM) and other lead compounds in the same assay, four analogs exhibited improved potency for the inhibition of SARS-CoV-2 viral replication in

Vero-E6 cells: EB46, EB54, EB55, and EB56, with EC$_{50}$ values of 2 μM, 2 μM, 3 μM, and 4 μM, respectively.

Since the high antiviral-EC$_{50}$/biochemical-IC$_{50}$ ratio observed in Vero E6 cells was found to be an artifact of the high efflux potential of Vero E6 cell line and may underestimate antiviral potency[18], we further evaluated these compounds in human Huh-7 cells which were transfected to overexpress ACE2 receptor (Huh-7$^{ACE2}$) before the addition of SARS-CoV-2 virus. Compared with GC376 (EC$_{50}$ = 51 nM) in the same assay, four of the analogs exhibited substantially improved antiviral potency: EB46, EB54, EB55, and NK01-63, with EC$_{50}$ values of 14 nM, 5 nM, 11 nM, and 6 nM, respectively (Supplementary Fig. 17b, Table 1). The improved antiviral-EC$_{50}$ values in human Huh-7$^{ACE2}$ cells for these analogs are not only in line with their nanomolar biochemical-IC$_{50}$ values, but also in agreement with their improvement over GC376 in biochemical inhibition potency. Considering both biochemical

inhibition and antiviral potency, EB46, EB54, and NK01-63 consistently exhibited outstanding improvements over GC376 for inhibiting SARS-CoV-2 in vitro and in cells. As a confirmatory assay, we further tested EB46, EB54, and NK01-63 in human Caco-2 cells against SARS-CoV-2 infection, where promising antiviral potency was consistently observed, with $EC_{90}$ values of 53 nM, 190 nM, and 81 nM, respectively.

For therapeutic consideration, we measured the cellular toxicity of all GC376 analogs in Huh-7$^{ACE2}$ cells (Supplementary Fig. 17c). As no compound exhibited significant toxicity at the tested concentrations up to 50 μM, which is approximately 10,000-fold more than the antiviral-$EC_{50}$ values of the most potent GC376 analogs in the same cells against SARS-CoV-2, the therapeutic potentials of optimized GC376 analogs appeared promising and selective.

To better understand the origin of the improvement in potency, we evaluated the binding of GC376 analogs to SARS-CoV-2 3CL protease. We confirmed that these analogs covalently bind to the active site cysteine 145 in the same manner as GC376, as incubation of 3CL protease with one equivalent of representative analogs EB48, EB56, or SL-4-241 blocked the covalent binding of 10 equivalents of compound 4 to the protease (Supplementary Fig. 17d). With the ITC assay, we confirmed that EB46, EB54, and EB55 bind to 3CL protease with 1:1 stoichiometry, indicating specificity for a single binding site (Supplementary Fig. 18). Furthermore, considering the covalent binding nature of GC376 analogs, we determined their $k_{inact}/K_I$ values in a kinetic enzymatic assay (Supplementary Fig. 19). The results suggested significant improvements of EB46 (3.3 min$^{-1}$ μM$^{-1}$), EB54 (5.3 min$^{-1}$ μM$^{-1}$), and NK01-63 (4.4 min$^{-1}$ μM$^{-1}$) in binding affinity over GC376 (1.3 min$^{-1}$ μM$^{-1}$), which is consistent with their improved inhibition potency (Table 1).

To further visualize the binding of GC376 analogs to the 3CL protease to understand the potency of analogs, we solved the co-crystal structures of 3CL with EB46, EB48, EB54, EB56, SL-4-241, NK01-14, NK01-48, and NK01-63 at 1.65 Å, 2.08 Å, 1.68 Å, 2.03 Å, 2.17 Å, 1.64 Å, 1.79 Å, and 1.55 Å, respectively (Fig. 4a–g and Supplementary Fig. 20).

Alignment of the 3CL-EB46 structure with the 3CL-GC376 structure that we solved, revealed that EB46 not only inherited all the interactions that GC376 featured with the surface of the protease (hydrogen bonds with Phe140, Gly143, His163, His164, and Glu166), but also exhibited an additional hydrogen bond between Asn142 of the protease and the m-Cl substituent on the benzyl ring (PDB ID:7JSU[37], Supplementary Fig. 21a). This additional interaction likely further stabilizes the inhibitor in the binding pocket and therefore increase the binding affinity. This unique feature of the m-Cl interaction explains the potency improvement of EB46 ($IC_{50} = 23$ nM) and NK01-57 (13 nM) over GC376 (35 nM) and NK01-14 (22 nM) in the biochemical assay, as GC376 and NK01-14 lacks the m-Cl in EB46 and NK01-57, which is the only structural difference between each pair of compounds.

Although a m-Cl moiety is also present in the structure of EB48, the cyclohexyl group of EB48 and SL-4-241, which replaced the isopropyl group of GC376, extended further into the hydrophobic P2 pocket and expanded the binding site into a more open state (Supplementary Fig. 21b). As this extension caused the loss of two hydrogen bonds between Q189 and the inhibitor backbone and moved the side chain of Asn142 away from the m-Cl of EB48, which should have formed a hydrogen bond, the cyclohexyl substitution in EB48 ($IC_{50} = 92$ nM) and SL-4-241 (75 nM) for the corresponding isopropyl group in EB46 (23 nM) and GC376 (35 nM) did not further improve the potency, which is the only structural difference between each pair of compounds (Supplementary Fig. 21c).

The impact of the cyclohexyl moiety is in contrast to a cyclopropyl substitution of the same isopropyl group, as in the co-crystal structure of 3CL-NK01-14 we found that a cyclopropyl moiety did not expand the P2 pocket (Supplementary Fig. 21d). Moreover, a cyclopropyl substitution in the structure of NK01-14 ($IC_{50} = 22$ nM), NK01-57 (13 nM), and NK01-69 (24 nM) consistently improved biochemical inhibition potency over the corresponding isopropyl compound GC376 (35 nM), EB46 (23 nM), and EB54 (29 nM). Furthermore, the combination of m-Cl and cyclopropyl in NK01-57 contributed to the highest biochemical inhibition potency among the tested analogs. However, since enhancement of GC376 analogs with cyclopropyl substitution over GC376 in the biochemical assay did not translate fully to enhanced potency in the cell-based viral inhibition assay, we concluded that the cyclopropyl moiety may impact permeability or activity in a cellular context. Nonetheless, the viability to tune GC376 activity via the modification of the isopropyl of GC376 was noted.

On the other hand, when we aligned co-crystal structures of all GC376 analogs with the 3CL-GC376 complex, we found the scaffold backbones beyond the benzyl moiety of all these potent inhibitors aligned well, which indicated not only a highly-specific recognition but also the feasibility to modify the benzyl group (Fig. 4i). Indeed, the top three GC376 analogs, EB46 (m-Cl), EB54 (o-Cl), and NK01-63 (m-CF$_3$), which exhibited outstanding improvements over GC376 for inhibiting SARS-CoV-2 in biochemical and cellular assays, have halogen substitutions on the benzyl group. While the m-Cl of EB46 exhibited additional polar interactions with Asn142 as noted above, two F atoms on the m-CF$_3$ of NK01-63 correspondingly established additional hydrogen bonds with Asn142. These polar interactions are absent in the co-crystal structures of GC376 and other less potent analogs bound to 3CL protease. The additional hydrogen bonds may further stabilize the inhibitors in the binding pocket and explain their significant improvements over GC376. Although the o-Cl moiety of EB54 and the trifluoro moiety of another potent inhibitor, NK01-48, did not directly show additional hydrogen bonding with Asn142 as with the m-Cl moiety of EB46 in the co-crystal structure (which might be due to the specific locations and properties of the halogens), the high anti-viral potencies of EB54 and NK01-48 indicated the potential of similar interactions during the dynamic process of binding. Additionally, alignment of the 3CL-EB56 structures with EB46 revealed that the naphthalene group of EB56 overlaps well with m-Cl benzyl group of EB46 and makes hydrophobic interactions with the lactam, which suggested an energetically-favored orientation and indicated available space in the binding pocket to accommodate more substituents beyond single halogens (Supplementary Fig. 21e). Therefore, we envision that additional modifications on the benzyl group may lead to stronger or additional hydrogen bonds with 3CL protease. Such modifications may even further improve the inhibitor potency.

Since these co-crystal structures confirmed that the bisulfite warhead of GC376 and its analogs is converted to an active aldehyde warhead in solution before reacting with the 3CL protease, the bisulfite form can be considered a prodrug of the active aldehyde form. To investigate the effect of the bisulfite warhead, we prepared EB33 and EB34, which are the aldehyde analogs of two representative GC376 analogs EB54 and EB46 (Supplementary Fig. 22a). We first confirmed that incubation of the 3CL protease with one equivalent of EB33 or EB34 blocked covalent binding of 10 equivalents of compound 4 to the protease after buffer exchange, which suggested that EB33 and EB34 covalently bind to the active site cysteine 145 without the bisulfite warheads (Supplementary Fig. 22b). We then found that EB33 ($IC_{50} = 10$ nM) and EB34 (9 nM), as the active aldehyde forms,

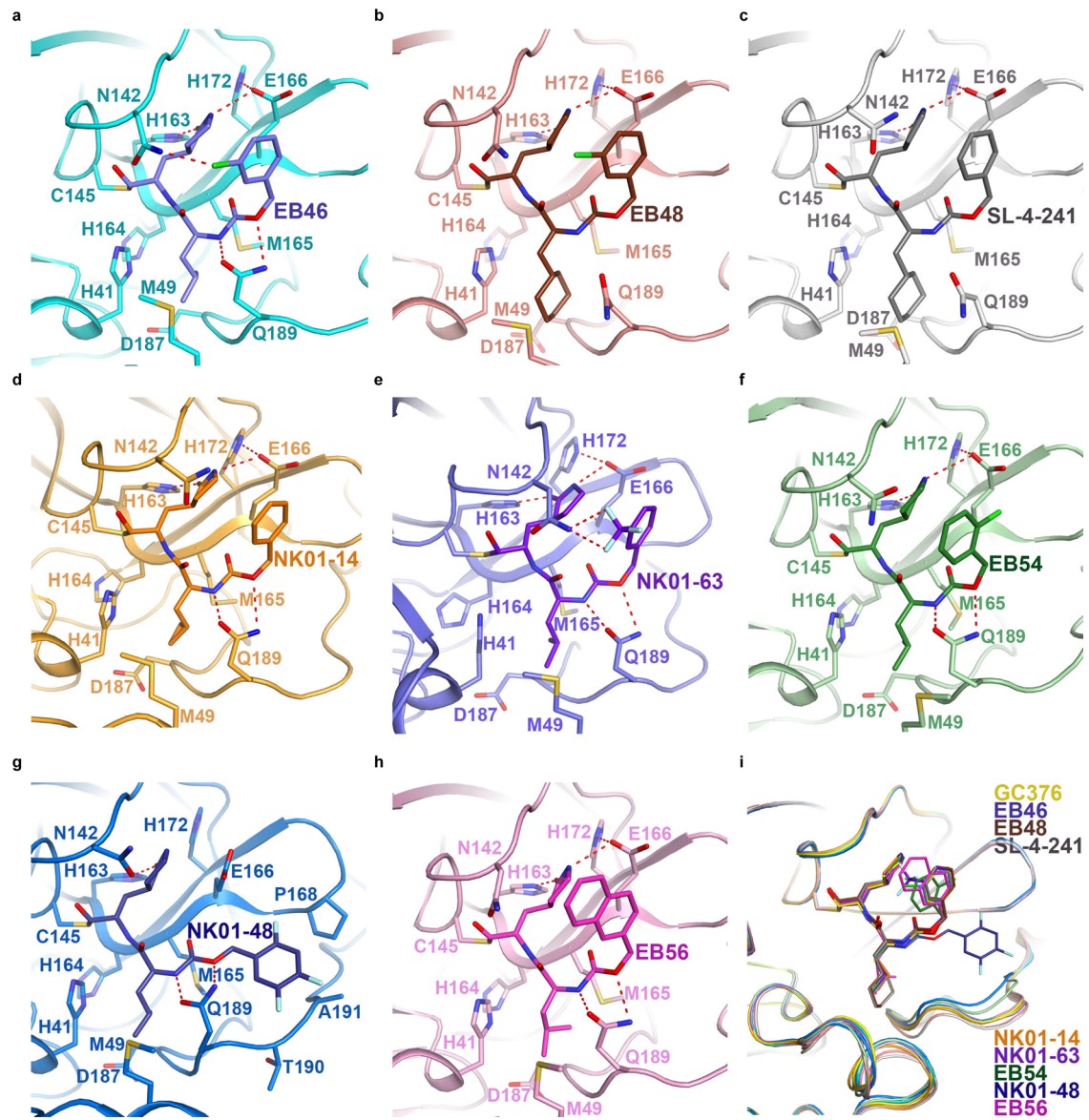

**Fig. 4 Co-crystal structures of GC376 analogs with SARS-CoV-2 3CL protease. a** The co-crystal structure of 3CL-EB46 complex. **b** The co-crystal structure of 3CL-EB48 complex. **c** The co-crystal structure of 3CL-SL-4-241 complex. **d** The co-crystal structure of 3CL-NK01-14 complex. **e** The co-crystal structure of 3CL-NK01-63 complex. **f** The co-crystal structure of 3CL-EB54 complex. **g** The co-crystal structure of 3CL-NK01-48 complex. **h** The co-crystal structure of 3CL-EB56 complex. **i** Alignment of the co-crystal structure of 3CL-EB46, 3CL-EB48, 3CL-EB54, 3CL-EB56, 3CL-SL4-241, 3CL-NK01-14, 3CL-NK01-48, 3CL-NK01-63, and 3CL-GC376 complex.

featured even higher biochemical inhibition potency than their corresponding bisulfite forms EB54 (29 nM) and EB46 (23 nM) against the 3CL protease, which might be due to bypassing the activation process in solution (Supplementary Fig. 22c). However, we also found that EB33 ($EC_{50} = 8$ nM) and EB34 (26 nM) exhibited lower antiviral potency than their corresponding bisulfite forms EB54 (5 nM) and EB46 (14 nM) against SARS-CoV-2 in Huh-7$^{ACE2}$ cells, suggesting the bisulfite warhead of GC376 and its analogs is important for their antiviral effect in a cellular context.

We next examined the binding selectivity of the optimized GC376 analogs for the viral protease over human proteases to evaluate potential off-target effects. First, we used EB33 and EB34 to represent the activated GC376 analogs in a cellular context. Accordingly, we found that GC376 analogs consistently exhibited selectivity towards the SARS-CoV-2 3CL protease over the human proteases chymotrypsin (Supplementary Fig. 22d). We

then evaluated the top GC376 analogs, EB54 and NK01-63, against a panel of additional human proteases (Supplementary Table 1). In general, EB54 and NK01-63 are highly selective for 3CL inhibition, displaying $IC_{50}$ values of >100 μM against many of the other human proteases and possessing only modest levels of inhibition of caspase 8 ($IC_{50} = 52$ μM and 18 μM), cathepsin B ($IC_{50} = 0.35$ μM and 1.1 μM), cathepsin K ($IC_{50} = 0.02$ μM and 0.04 μM), and cathepsin S ($IC_{50} = 0.006$ μM and 0.03 μM). High potency inhibition was observed, however, of cathepsin L for both compounds ($IC_{50} = 0.001$ μM and 0.006 μM). Cathepsin L was recently reported to play an important role in SARS-CoV-2 viral entry by activating the viral spike protein in the endosome or lysosome[38–41]. Since studies have indicated that cathepsin L inhibitors can block or substantially decrease SARS-CoV-2 viral entry without showing toxicity to the host, these optimized GC376 analogs are expected to effectively block SARS-CoV-2 infection and replication via dual inhibition of 3CL protease and

cathepsin L[39,40,42–44], and may thus act as multi-targeted anti-virals. Meanwhile, from another perspective, there is an emerging hypothetical concern on the development suitability of anti-virals working through cathepsin L because the virus may adopt alternative entry pathways, such as TMPRSS2[45]. However, NK01-63 and EB54 ($IC_{50} = 6$ nM and 1 nM, respectively) were less active than GC376 ($IC_{50} = 0.33$ nM[46]) on the inhibition of human cathepsin L. But they are more active than GC376 on the inhibition of SARS-CoV-2 3CL protease and suppression of SARS-CoV-2 replication. This suggested NK01-63 and EB54 are more likely to primarily work through 3CL inhibition in SARS-CoV-2 cellular antiviral assays.

To further confirm this, we evaluated these compounds in human Huh-7 cells which were transduced to overexpress both ACE2 receptor and TMPRSS2 (Huh-7[ACE2+TMPRSS2]) before the addition of SARS-CoV-2 virus. Compared with GC376 ($EC_{50} = 146$ nM) in the same assay, EB46, EB54, and NK01-63 consistently exhibited improved antiviral potency, with $EC_{50}$ values of 47 nM, 28 nM, and 146 nM, respectively (Supplementary Fig. 22e). Considering that EB54 and NK01-63 exhibited no inhibition of TMPRSS2 in the protease panel, we concluded that even in the scenario of redundant expression of viral entry pathways, the improved GC376 analogs that we identified still can effectively inhibit SARS-CoV-2 within a cellular context.

Finally, since the identification of GC376 as the lead compound was based on the conservation of 3CL protease substrate-binding pockets across coronaviruses, we expected GC376 and its analogs to be effective against other coronaviruses. To verify this hypothesis, we transfected HEK293T cells with 3CL proteases from different coronaviruses to induce cytotoxicity so that inhibition of the corresponding 3CL protease can protect the cells[34]. Accordingly, we found that the optimized GC376 analogs, EB46, EB54, and EB56, not only featured improved potency over GC376 to inhibit the activity of the SARS-CoV-2 3CL protease, but also exhibited better potencies than GC376 against the original SARS-CoV 3CL protease and equivalent capabilities to inhibit the MERS-CoV 3CL protease in cells, suggesting pan-3CL-protease activity of these inhibitors (Fig. 5a). Additionally, this cell-based inhibition assay confirmed that these compounds can effectively inhibit the tested viral proteases within a cellular context. To further verify the pan-coronavirus activity of these inhibitors, we tested them in cells infected with a panel of viruses, where we found EB46, EB54, and NK01-63 can effectively block the viral replication of human coronavirus alpha OC43 and human coronavirus beta 229E, with $EC_{50}$ values < 100 nM in Huh-7 human cells (Fig. 5b). They can also block replication of MERS-CoV ($EC_{50}$ values < 1 μM) and SARS-CoV ($EC_{50}$ values < 3 μM) in Vero 76 cells. Their activity appeared to be specific towards coronavirus, as no effect was observed on the replication of Influenza H1N1, which also indicates the compounds are not non-specific inhibitors of viral entry. In conclusion, these data suggest pan-coronavirus activity of these inhibitors towards currently existing and, more importantly, potentially future coronaviruses.

**In vivo toxicity and pharmacokinetic study of NK01-63.** Considering all of the above properties of the GC376 analogs that we developed, we nominated NK01-63 as the most promising compound for in vivo studies. We first monitored the body weight change of C57BL/6 mice treated with 20 mg/kg NK01-63 or water vehicle via intraperitoneal (IP) or oral (PO) dose once per day for 14 consecutive days. No significant change in body weight was observed as compared to the vehicle group, showing that NK01-63 has no in vivo toxicity via either route of administration (Fig. 6a). In a separate pharmacokinetic study, we measured the molar

concentration of NK01-63 in plasma and lung of C57BL/6 mouse after treatments with 20 mg/kg NK01-63 via IP or PO dose (Fig. 6b, c). With the $EC_{90}$ value of NK01-63 (81 nM) determined from the Caco-2 cell-based SARS-CoV-2 anti-viral assay as a reference, we found both routes of administration can deliver a substantial concentration of NK01-63 to plasma (IP: 39 μM, 486 × $EC_{90}$; PO: 1.9 μM, 24 × $EC_{90}$) and lung (IP: 15 μM, 181 × $EC_{90}$; PO: 1 μM, 12 × $EC_{90}$). Although the half-life of NK01-63 in plasma is shorter (IP: $T_{1/2} = 0.4$ h; PO: $T_{1/2} = 0.8$ h), its half-life in critical tissues such as lung is long (IP: $T_{1/2} = 2.2$ h; PO: $T_{1/2} = 3.4$ h). Therefore, the concentrations of NK01-63 in lung at 24 h after treatment are still above its $EC_{90}$ value (IP: 313 nM, 3.9 × $EC_{90}$; PO: 110 nM, 1.4 × $EC_{90}$). These in vivo data suggested the promising in vitro anti-viral potency of NK01-63 is likely to be consistently translated into the scenario of clinical applications against COVID-19.

**Discussion**

Together, these data demonstrate the discovery of a series of potent small-molecule GC376-based inhibitors for SARS-CoV-2 3CL protease with drug-like properties. In light of the promising results of in vivo studies on GC376 and other 3CL inhibitors[22,27,47–49], we envision the GC376 analogs that we identified with improved potency will contribute to the development of effective COVID-19 treatments. Considering all of the above properties of these compounds, we nominated NK01-63 as the most promising compound and have termed it *coronastat*. Coronastat features the most outstanding potency among all reported SARS-CoV-2 3CL protease inhibitors currently in development (Supplementary Table 2). We propose performing IND (Investigational New Drug)-enabling preclinical studies on coronastat to evaluate its potential for clinical development. Additionally, since a recent study revealed that 3CL protease inhibitors have additive/synergistic activity in combination with anti-viral drug Remdesivir[49], combination treatments of coronastat with other first line anti-virals might further boost its therapeutic efficacy.

In our study, we initially used Vero cells to evaluate the cellular antiviral potency ($EC_{50}$) of 3CL inhibitors. However, a recent study revealed that the high antiviral-$EC_{50}$/biochemical-$IC_{50}$ ratio observed in Vero cells was an artifact of the high efflux potential of the Vero cell line and may underestimate the antiviral potency in human lung cells, the relevant tissue for COVID-19[18]. Accordingly, we used the Huh-7[ACE2] cell line, which lacks drug efflux, for the evaluation of antiviral potency during the later lead optimization stage. Since Huh-7 cells were engineered to express human ACE2 receptor, it can be infected with SARS-CoV-2. Aiming to rule out any potential interference of the cell engineering process, as a confirmatory assay, we also tested optimized lead compounds in human epithelial Caco-2 cells against SARS-CoV-2 infection. We found the improvement of the analogs over GC376 was consistently validated in all types of cells that we used.

In this study, we applied ITC to investigate the binding stoichiometry of inhibitors to the 3CL protease, which enabled the selection of compounds with specificity for a single binding site. However, since dissociation constants ($K_d$) derived from ITC data were not generally suitable for assessing covalent inhibitors, we instead evaluated GC376 analogs with $k_{inact}/K_I$ values, which describe the efficiency of covalent bond formation after the initial reversible binding. We indeed observed a better correlation between compound potency and $k_{inact}/K_I$ values, rather than $K_d$ values derived from ITC data.

It is noteworthy that during the revision stage of this manuscript, another research group who also works on the development of GC376 analogs independently reported the chemical structures of EB34, NK01-14, and NK01-57 as 3CL protease

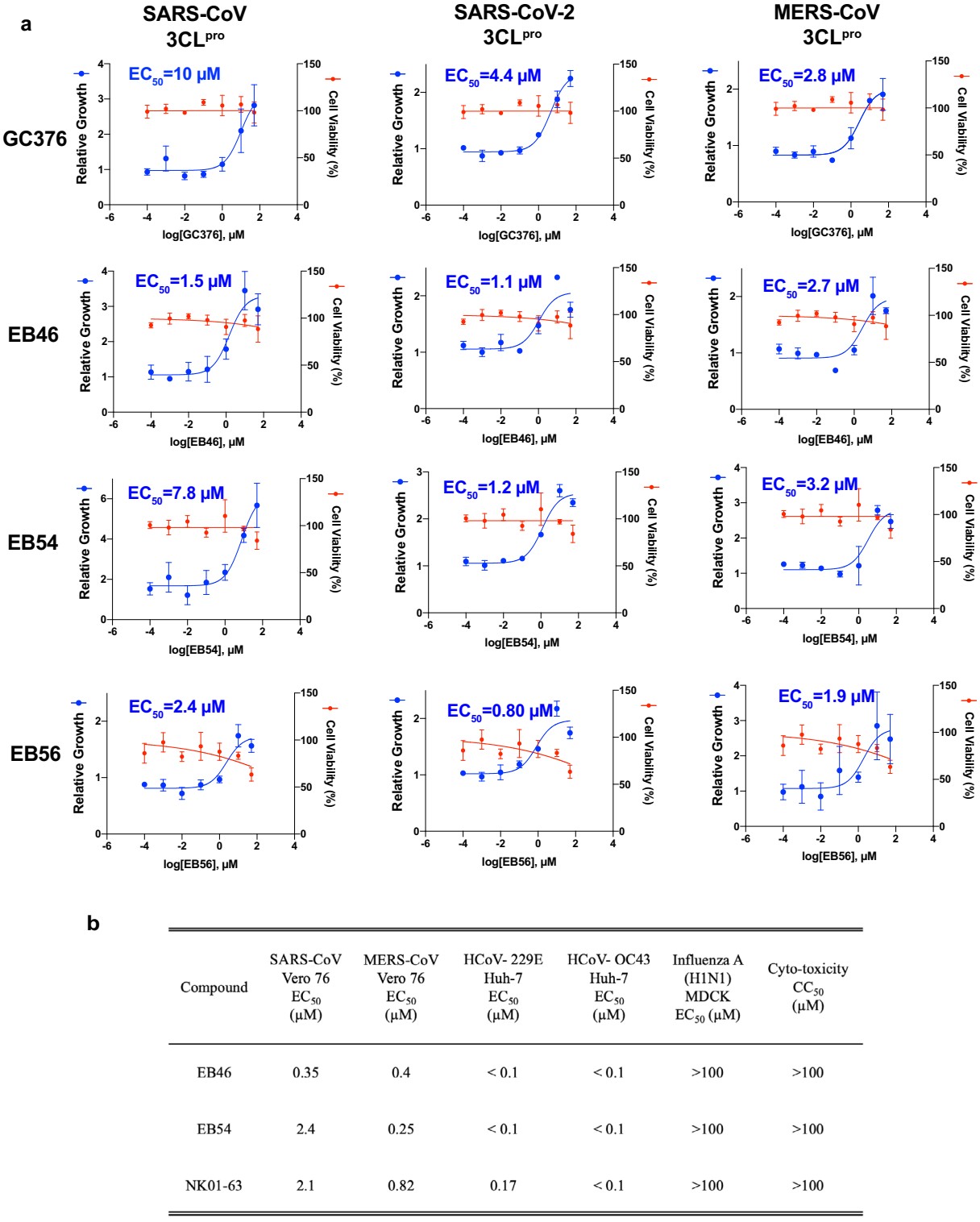

**Fig. 5 Pan-coronavirus inhibition potency of optimized GC376 analogs. a** Evaluations of GC376 analogs in a cell-based SARS-CoV, SARS-CoV-2, and MERS-CoV 3CL protease inhibitor assay. Expression of viral 3CL protease suppressed the viability of the 293T cells. Inhibitors of the protease rescue cell growth and boost the amount of crystal violet cell staining which is read on an absorbance plate reader. Cells expressing EYFP were included as control. Experiments were performed in quartet. Data are plotted as the mean ± s.d., $n = 4$ biological replicates. Source data are provided as a Source Data file. **b** Ability of EB46, EB54, and NK01-63 to inhibit viral infection in cell culture.

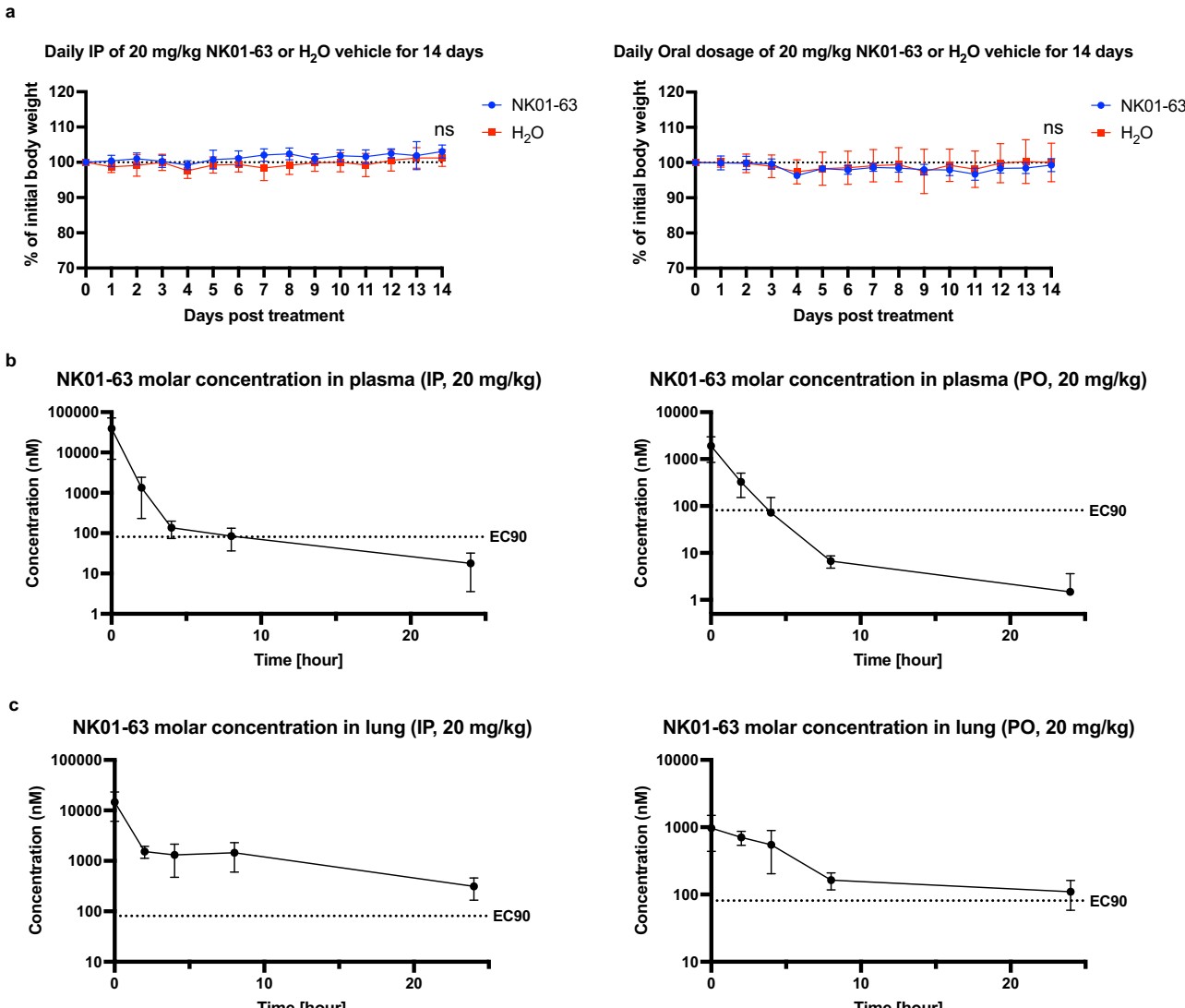

**Fig. 6 In vivo toxicity and pharmacokinetic study of NK01-63. a** Body weight change of C57BL/6 mouse treated with 20 mg/kg NK01-63 or water vehicle via intraperitoneal (IP) or oral (PO) dose for 14 consecutive days on an in vivo toxicity study. Data are plotted as the mean ± s.d., $n = 10$ biological replicates. Two-tailed $t$ test were performed on weight of 14 days post initial treatment. $P^{ns}$ of IP = 0.07, $P^{ns}$ of PO = 0.68. **b**, **c** Molar concentration of NK01-63 in plasma or lung of C57BL/6 mouse 0 h, 2 h, 4 h, 8 h, or 24 h after treatment with 20 mg/kg NK01-63 via intraperitoneal (IP) or oral (PO) dose. Data are plotted as the mean ± s.d., $n = 8$ biological replicates. $EC_{90}$ value of NK01-63 is from Caco-2 cell-based SARS-CoV-2 antiviral assay. Source data are provided as a Source Data file.

inhibitors[24]. In their report, NK01-57 was the top compound. Indeed, their work supports the effectiveness of our optimization strategy reported in this manuscript, as the inhibitors we describe (including coronastat) showed further improvement over NK01-57 in a side-by-side comparison in the $k_{inact}/K_I$ enzymatic assays and cellular anti-viral experiments. We envision that our research efforts, jointly with other works of the field, will eventually lead us to a suite of effective small molecular anti-viral treatments for COVID-19.

In this study, not only did we identify potent drug-like inhibitors for SARS-CoV-2 3CL protease, but we also presented a roadmap for rapid discovery of inhibitors for novel viruses, and evaluating their suitability for development. Based on the conservation of this target and its similarity with known proteins, we repurposed previously-reported inhibitors as starting inhibitor candidates. After a thorough examination of the inhibitory effect of the candidates on the target in vitro and in cells, with a consideration of their selectivity and metabolic stability, we prioritized GC376 for further

development. Given that the compound GC376 is water-soluble, safe, and effective in a feline study to treat coronavirus infections, we would expect the best analogs of GC376 to inherit these merits[50], which will also be advantages of these inhibitors over other inhibitor candidates reported recently[13,14,16]. Additionally, based on the observation that the 3CL protease substrate-binding pocket is highly conserved across different coronaviruses, we expect coronastat to have potential effectiveness not only on all variants of SARS-CoV-2 3CL protease, but also against other coronaviruses, including a potential SARS-CoV-3 in the future.

## Methods

**Preparation and synthesis of compounds**. GC376 was purchased from Aobious, Ebselen was purchased from Cayman Chemical, and MAC-5576 was purchased from Maybridge. MAC5576 analogs were purchased from Sigma Aldrich (T71: T311871) and Enamine (Z14: Z2239052914, Z97: Z2239051897, Z21: Z1245217921, Z65: Z1291412865, Z26: Z1245218326, Z91: Z1291412791, Z50: Z1245218850, Z32: Z1245218632, Z43: Z1216060043, and Z61: Z1216059461). Ebselen analog AZVIII-40A (1,2-Benzisothiazol-3(2H)-one) was purchased

from Alfa Aesar. Synthesis of all other compounds was summarized in Supplemental Note 1.

**Expression and purification of SARS-CoV-2 3CL protease.** The SARS-CoV-2 3CL protease gene was codon optimized for bacterial expression and synthesized (Twist Bioscience), then cloned into a bacterial expression vector (pGEX-5X-3, GE) that expresses the protease as a fusion construct with a N-terminal GST tag, followed by a Factor Xa cleavage site. After confirmation by Sanger sequencing, the construct was transformed into BL21 (DE3) cells. *E. coli* were inoculated and grown overnight as starter cultures, then used to inoculate larger cultures at a 1:100 dilution, which were then grown at 37 °C, 220 RPM until the OD reached 0.6–0.7. Expression of the protease was induced with the addition of 0.5 mM IPTG, and then the cultures were incubated at 16 °C, 180 RPM for 10 h. Cells were pelleted at 4300 × *g* for 15 min at 4 °C, resuspended in lysis buffer (20 mM Tris-HCl, pH 8.0, 300 mM NaCl), homogenized by sonication, clarified by centrifuging at 25,000 × *g* for 1 h at 4 °C. The supernatant was mixed with Glutathione Sepharose resin (Sigma) and placed on a rotator for 2 h at 4 °C. The resin was then repeatedly washed by centrifugation at 2600 × *g* for 15 min at 4 °C, discarding of the supernatant, and then resuspension of the resin in fresh lysis buffer. After ten washes, the resin was resuspended in lysis buffer, and Factor Xa was added and incubated for 18 h at 4 °C on a rotator. The resin was centrifuged at 2600 × *g* for 15 min at 4 °C, and then the supernatant was collected and concentrated using a 10 kDa concentrator (Amicon) before being loaded onto a Superdex 10/300 GL column for further purification by size exclusion chromatography. The appropriate fractions were collected and pooled with a 10 kDa concentrator, and then the final product was assessed for quality by SDS-PAGE and measurement of biochemical activity.

**ITC binding assay.** All of the stock ligands were dissolved in 100% DMSO with the exception of GC376, which was dissolved in water. The final buffer formulation was 50 mM Tris, 1 mM EDTA, 1% glycerol, and 0.8–1% DMSO. DMSO was added to the water-soluble ligands to avoid any potential differences due to the presence of an organic solvent. 3CL-protease was diluted to a concentration between 5 and 14 μM with 50 mM Tris buffer pH 7.5 and 0.8–1% DMSO. All samples and buffer were degassed for 10 min prior to loading on the instrument. Solutions were thawed the day of use and kept at 4 °C between each duplicate run.

Titrations were performed in an Affinity ITC LV and a Nano ITC LV (TA Instruments-Waters Corporation), with an active cell volume of 182 μL and 170 μL, respectively, and a load volume of 300 μL for both. The cell was loaded with 3CL protease and the syringe was loaded separately with each small molecule ligand. The systems were stirred at their optimal stir rate: 150 rpm for the Affinity and 350 rpm for the Nano. Incremental injections of 2 μL were delivered every 250 s or less; the spacing between each injection was controlled by the software, ITCRUN® where baseline resolution was determined realtime with a criteria of 0.1 μW/h and 0.01 μW prior to the next injection. Prior to the first injection, the system was equilibrated to a slope of less than 0.3 μW/h and less than 0.01 μW standard error. Data is plotted in joules and exothermic events are downward peaks in the raw thermogram.

Each ligand was tested in duplicate. The appropriate negative control of small molecule ligand into the running buffer was also performed. Ultimately, the heat of dilution that was used to baseline resolve the data was the heat of dilution after saturation.

All data were fitted using NanoAnalyze®. The exact concentrations were loaded into the software for data normalization and least-squares fitting within the software. The data were fitted with a one-site model that fits the parameters of stoichiometry (*n*), enthalpy (Δ*H*), and affinity (*K*_a). The values for *K*_d, Δ*G*, and Δ*S* are mathematically determined from the fit parameters. The exception to fitting was the ebselen sample, which was fitted to a multiple site model that assumes two binding sites with unfixed stoichiometry. The heat of dilution used to correct the data prior to fitting was the average area of the last few injections, which agreed within 1 μJ of the heat associated with the negative control of ligand into buffer. Another exception is for EB46, where data point #4 on one set of data deviated from the other points leading to a large error and uncertainty in *K*_d. Data point #4 was therefore removed before data fitting due to the possibility of an air bubble being released concurrent with the end of the injection.

**Measurement of SARS-CoV-2 3CL protease inhibition.** The in vitro biochemical activity of the SARS-CoV-2 3CL protease was measured as previously described[14]. Practically, the fluorogenic peptide MCA-AVLQSGFR-Lys(DNP)-Lys-NH2, corresponding to the nsp4/nsp5 cleavage site in the virus, was synthesized (GL Biochem), then resuspended in DMSO to use as the substrate.

For the measurement of IC_50 values, in a 96 well plate, the protease in the assay buffer (50 mM Tris-HCl, pH 7.5, 1 mM EDTA) was first added to each well at a final concentration of 0.2 μM. Serial dilutions of the test compound were prepared in the assay buffer. The substrate was then added at 20 μM, and then fluorescence (Excitation 320 nm, Emission 405 nm) at 30 °C was continuously measured on a plate reader for 20 min. Inhibition was then calculated by comparison to control wells with no inhibitor added (negative control) and with no protease added (positive control). IC_50 values were determined by fitting data with Log(inhibitor) vs. normalized response curve (the standard inhibition curve, GraphPad Prism).

**Determination of *k*_inact/*K*_i values.** The determination of *k*_inact/*K*_i values for GC376 and its analogs was performed by WuXi AppTec following literature methodology[51,52]: Frist, IC_50 values of test compounds were determined in SARS-CoV-2 3CL protease enzymatic assay. Compounds were 3-fold serially diluted for 10 doses starting at 10 μM as the top concentration and added to assay plates, in duplicate wells. 25 nM of SARS-CoV-2 3CL protease was added to the assay plate containing the compounds, and incubated for 30 min at room temperature. For fluorescence controls, no SARS-CoV-2 3CL protease but assay buffer was added instead. 25 μM of the substrate (MCA-AVLQSGFR-Lys(DNP)-Lys-NH2) was added to the assay plate, and incubated for 60 min at 30 °C. Then the fluorescence signal will be detected using a microplate reader. IC_50 values of the compounds were then calculated with the GraphPad Prism software. Then, for the determination of the *k*_inact/*K*_I values of GC376 and its analogs in SARS-CoV-2 3CL protease enzymatic assay, compounds were 1.5-fold serially diluted for 7 doses starting at 4.5 × IC_50 value and added to assay plates, in triplicate wells. 25 μM of substrate and 25 nM of 3CL protease was added to assay plates containing compounds. The assay plates were incubated at 30 °C in a plate reader and fluorescence signals were read every 2 min. For each concentration of a compound, a kinetic curve was generated with the fluorescence readout to get reaction velocity at each time point. To determine *k*_obs values (units: min^−1), the decrease in the natural logarithm of the activity over time was plotted for each inactivator concentration, and *k*_obs values were described as the negative slopes of the lines. This is based on Eq. (1):

$$\text{Remaining activity}\% = 100 * e^{(-k_{obs}*\text{time})} \tag{1}$$

This yields the *k*_obs, an observed first-order rate constant with units of inverse time, at each inhibitor concentration. The *k*_obs values are then fit to the following Eq. (2) to determine the *k*_inact and *K*_I:

$$k_{obs} = \frac{k_{inact}[\text{Inhibitor}]}{K_I + [\text{Inhibitor}]} \tag{2}$$

All fittings were performed in GraphPad Prism. The *k*_inact/*K*_I values of each inhibitor were then calculated.

**Chymotrypsin activity assay.** The Panc-1 cells used in this study were purchased from ATCC. Cells were maintained at 37 °C in a humidified atmosphere with 5% CO_2. Panc-1 cells were grown in Dulbecco's Modified Eagle Medium (DMEM, Invitrogen) which was supplemented with 10% fetal bovine serum (Gibco) and penicillin-streptomycin (Invitrogen).

The in vitro biochemical activity of human chymotrypsin was measured using the Chymotrypsin Assay Kit (Fluorometric) (Abcam, ab234051). Panc-1 cells were cultured, collected and homogenized on ice with the chymotrypsin assay buffer provided by the kit. The protein concentration of the Panc-1 cell lysate was determined with Pierce BCA protein assay kit (Thermo Scientific, #23225). In a 96 well plate, Panc-1 cell lysate of 100 μg total protein was first added to each well. 10 μM or serial dilutions (if inhibition of the chymotrypsin activity was observed at 10 μM) of the test compound were prepared in the assay buffer, and then incubated with the lysate for 10 min on ice. Reaction mixtures comprised of chymotrypsin activator and fluorogenic substrate were then added to make a total volume of 100 μL per well. We then started recording the fluorescence (*E*_x/*E*_m = 380/460 nm) in kinetic mode for 30 min at 25 °C to record the linear reaction phase. Inhibition was then calculated by comparison to control wells with no inhibitor added (negative control) and with no lysate added (positive control). IC50 values were determined by fitting data with Log(inhibitor) vs. normalized response curve (the standard inhibition curve, GraphPad Prism).

**Human protease selectivity panel.** Protease assays were performed by Reaction Biology Corp (Malvern, PA, USA). The source of protease, substrate and its concentration, excitation and emission wavelengths, reaction buffer, and control inhibitors were summarized in Supplementary Table 3. Particularly, the following buffers were correspondingly applied: Buffer A (25 mM Tris pH 8.0, 100 mM NaCl, 0.01% Brij35), buffer A′ (50 mM Tris pH 8.0, 150 mM NaCl, 0.005% Brij35), buffer B′ (25 mM MES pH 6, 50 mM NaCl, 0.005% Brij35, 5 mM DTT), buffer B + EDTA (75 mM Tris pH 7.0, 1 mM EDTA, 0.005% Brij35, 3 mM DTT), buffer L (400 mM NaAcetate pH 5.5, 4 mM EDTA, 8 mM DTT), ACE2 buffer (75 mM Tris, pH 8.5, 1 M NaCl), Elastase buffer (50 mM Tris, pH 7.5, 1 M NaCl, 0.05% Brij35), 1X Caspase buffer 2 (50 mM HEPES pH7.4, 1 M Na Citrate, 100 mM NaCl, 0.01% CHAPS, 0.1 mM EDTA, 10 mM DTT), MMP Buffer (50 mM HEPES pH 7.5, 10 mM CaCl_2, 0.01% Brij-35, stored at 4 °C and added 0.1 mg/ml BSA in buffer before use), and Furin buffer (100 mM Tris HCl pH 7.5, 1 mM CaCl_2, 0.005%Brij35).

In particular for cathepsin L, two substrates were used to explore the suitable substrate: Z-FR-AMC and MCA-PLGL-Dap(Dnp)-AR-NH2. The method optimization demonstrated cathepsin L is more active on Z-FR-AMC as substrate as compared to MCA-PLGL-Dap(Dnp)-AR-NH2. Therefore, when performing the IC50 assays, 0.5 nM Cathepsin L was used with MCA-PLGL-Dap(Dnp)-AR-NH2 substrate, while 0.042 nM Cathepsin L was used with Z-FR-AMC substrate. As a result of higher concentration of Cathepsin L being used in the assay, IC50 values of EB54 and NK01-63 increased (as shown in the Supplementary Table 1). The general protocol for all proteases assays is: First, 2X enzyme or buffer as no enzyme control was delivered into the wells. Compounds (in 100% DMSO) or

DMSO as no compound control were then delivered into the wells using Acoustic technology (Echo550; nanoliter range). After an incubation of 20 min, 2X substrate was delivered to initiate the reaction. After spin and shake, measurement by EnVision plate reader was started at room temperature, for 25 measurements with 5 min interval (total measurement time is 2 h). Data were analyzed by taking the slope (signal/time) of the linear portion of the measurement. The slope was calculated using Excel, and curve fits performed using Prism software.

**Mass spectrometry analysis of protein-inhibitor complex.** To directly demonstrate the binding of covalent inhibitors, 50 μM of the 3CL protease was incubated with 500 μM of inhibitor in a buffer comprised of 50 mM Tris-HCl (pH 7.5), 1 mM EDTA for 1 h at 4 °C to acquire the protein-inhibitor complex before analysis by Mass Spectrometry. To indirectly demonstrate the binding of reversible covalent inhibitors, the binding of which might be otherwise sensitive to the low pH and other conditions of MALDI and RPLC-MS analysis protocol, 50 μM of the 3CL protease was first incubated with DMSO or 50 μM (1 equivalent) reversible inhibitors in a buffer comprised of 50 mM Tris-HCl (pH 7.5), 1 mM EDTA for 1 h at 4 °C. After a complete buffer exchange to remove any residual free inhibitor, 500 μM of compound 4 (10 equivalent) was then added to the 3CL protease, where compound 4 will modify 3CL without inhibitor bound at cysteine 145, but not 3CL that had already covalently reacted with the pre-incubated inhibitors at cysteine 145.

For MALDI-TOF MS analysis, 1 μL of the protein-inhibitor complex (3CL only, 3CL-compound 4, 3CL-MAC-5576, 3CL-Z43, (3CL+EB33)-compound 4, (3CL +EB34)-compound 4, (3CL+EB48)-compound 4, (3CL+EB56)-compound 4, and (3CL+SL-4-241)-compound 4, respectively) was mixed with 9 μL of 10 mg/ml sinapinic acid in the matrix solution (70:30 water/acetonitrile, with 0.1% TFA). 1.0 μL of the final mix was deposited onto the target carrier and allowed to air dry. All MALDI spectra of the protein-inhibitor complex were compared with ligand-free 3CL to determine the mass shift. MALDI spectra were collected with the software flexControl, while MALDI data were analyzed by flexAnalysis.

For LC-MS/MS analysis of intact and digested protein-inhibitor complex, 10 μL of 3CL or 3CL-inhibitor complex (2 mg/mL) was denatured with 90 μL 8 M guanidine-HCl and 5 mM DTT in 50 mM Tris-HCl at pH 7.6, for 30 min before being alkylated with 15 mM iodoacetamide for 20 min in the dark. Zeba$^{TM}$ desalting column (0.5 mL, Thermo Fisher) was used to desalt the protein. 0.5 μg of chymotrypsin (Promega, sequencing grade) was added to each sample and digestion was conducted for 1 hr at 37 °C. To quench the reaction, formic acid was added to each sample to a final concentration of 1%. The digested protein samples were submitted for LC-MS/MS analysis. Briefly, 10 μL of each sample in QuanRecovery with MaxPeak 96 well sample plate (Waters) was injected into the BioAccord system (Waters). An ACQUITY UPLC BEH C$_{18}$ 130 Å, 1.7 μm, 2.1 mm × 100 mm column was used for peptide separation. A linear gradient of 99% A (mobile phase A: 0.1% formic acid in milli Q water, mobile phase B: 0.1% formic acid in acetonitrile) to 60% A over 60 min was utilized at 0.25 mL/min flow rate. The RDa Detector was operated in full scan with fragmentation mode (2 Hz) with data acquired over a mass range of 50–2000 $m/z$. Data were analyzed through UNIFI with 10 ppm as the mass tolerance and minimal three fragment ions were used to assign the peptide.

For Intact protein LC-MS analysis to determine compound 4 binding and to indirectly observe GC376/GC373 binding (via reactions with compound 4 or iodoacetamide) with 3CL, reversed-phase LC-ESI-MS (RPLC-MS) of intact protein-inhibitor complex (denaturing) was conducted. It was performed on a Waters H-Class Plus UPLC connected to a Xevo G2-XS QTOF mass spectrometer running in ESI+ mode. The method used a Waters Acquity UPLC Protein BEH C4 (2.1 × 100 mm, 1.7 μm) column with a gradient of 5–95% B (A: water + 0.2% formic acid, B: acetonitrile + 0.2% formic acid) at a flow rate of 0.3 mL/min. The binding of 3CL with ebselen was carried out on a Waters BioAccord$^{TM}$ system with a similar gradient of 5–95% B (A: water + 0.1% formic acid, B: acetonitrile + 0.1% formic acid) at a flow rate of 0.4 mL/min. The LC-MS system was controlled by the software MassLynx, while the software UNIFI$^{TM}$ was used to process the MS deconvolution through MaxEnt 1.

Native size exclusion chromatography-ESI-MS (SEC-MS) of intact protein-inhibitor complex (Non-denaturing) was conducted to directly observe 3CL binding with GC376 and ebselen. It was performed on a Waters I-Class Plus UPLC system connected to a RDa mass detector (BioAccord system). The method used a Waters Acquity UPLC Protein BEH SEC (2.1 × 150 mm, 1.7 μm, 200 Å) column with an isocratic flow of 50 mM ammonium acetate (pH = 6.8) at a flow rate of 0.1 mL/min. The parameters used on RDa Detector: cone voltage 30 V, capillary voltage 1.5 kV, desolvation temperature 350 °C. The SEC-MS system was controlled by the software MassLynx, while the software MS deconvolution was completed through UNIFI using MaxEnt 1.

**Plasma and microsome stability assays.** Liver microsomes (Human or CD-1 mouse) were diluted to 0.5 mg/mL in a solution containing 100 mM PBS buffer at pH 7.4, an NADPH regenerating system (Promega), and test compounds and control compounds were diluted from DMSO stock solutions to 1 μM, unless otherwise indicated. Control reactions were made by withholding the addition of the NADPH regenerating solution, which enabled us to assess whether compound

stability was the result of NADPH-dependent metabolism. The mixture was incubated at 37 °C under gentle rotation and quenched by aliquoting 10 μL of solution into 90 μL ice-cold acetonitrile containing an internal standard. To evaluate the plasma stability of lead compounds, plasma (Human or CD-1 mouse) was warmed to 37 °C in a water bath, and test compounds were added to the plasma at 10 μM. Plasma samples were incubated at 37 °C under gentle rotation. At indicated time points, samples were quenched by aliquoting 10 μL of solution into 90 μL ice-cold acetonitrile containing an internal standard. Samples were collected and analyzed by LC-MS on a Waters H-Class Plus UPLC connected to a Xevo G2-XS QTOF mass spectrometer running in ESI+ mode. The method used a Waters Acquity UPLC BEH C18 (2.1 × 50 mm, 1.7 μm) column with a gradient of 23–100% B (A: water + 0.2% formic acid, B: 4:1 methanol/isopropanol).

**Metabolite identification.** Compound 4 metabolite discovery and identification experiments were performed on a Waters I-Class Plus UPLC connected to a Synapt G2-Si QTOF mass spectrometer with ion mobility. The method used a Waters Acquity UPLC CSH C18 (2.1 × 100 mm, 1.7 μm) column with a gradient of 25–81% B (A: water + 0.2% formic acid, B: 4:1 methanol/isopropanol). Traveling wave IMS was used with a variable wave velocity from 800 to 400 m/s. Metabolite discovery was carried out using ESI+ in HDMSe mode with a collision energy ramp from 55 to 75 eV in the high energy function. The data were processed using the metabolite discovery workflow in Waters' UNIFI software. This was supplemented with directed LC-MS/MS experiments to confirm metabolite structures and sites of oxidation.

**Solubility Measurements.** Solubility measurement samples were prepared in a 96-well Corning Costar plate with a total sample volume of 200 μL in either PBS buffer with 1.0% DMSO (v/v) or pure water. A BMG Labtech NEPHELOstar nephelometer was used to measure turbidity of the samples with the following parameters per the instrument manufacturer: Gain = 80, Laser Focus = 2.50 mm, and Laser Intensity = 90%. Samples were shaken in the nephelometer by orbital shaking for 3 s prior to the turbidity measurement. Sample values were blanked in the instrument software using the corresponding PBS and DMSO condition values. Turbidity 4000 NTU Calibration Standard Formazin (Millipore-Sigma) was used as a positive control for all turbidity measurements. The turbidity of each sample was measured within 20 min of initial preparation.

**Mammalian cell-based assay to identify coronavirus protease inhibitor (SARS-CoV, SARS-CoV-2, and MERS).** The HEK293T cells used in this study were purchased from ATCC. Cells were maintained at 37 °C in a humidified atmosphere with 5% CO$_2$. HEK293T cells were grown in Dulbecco's Modified Eagle Medium (DMEM, Invitrogen) which was supplemented with 10% fetal bovine serum (Gibco) and penicillin-streptomycin (Invitrogen). HEK293T cells were confirmed to be free of mycoplasma contamination with the Agilent Myco-Sensor PCR Assay Kit prior to use.

3CL protease expression constructs were cloned into the pLEX307 backbone (Addgene #41392) using Gateway LR II Clonase Enzyme mix (Invitrogen). Gene fragments containing the SARS-CoV, SARS-CoV-2, or MERS 3CL proteases used in this study were ordered from Twist Biosciences. Plasmid DNA was transformed into NEB 10-beta high-efficiency competent cells. Sanger sequencing to verify proper inserts was done for all plasmids used in this study (Genewiz). Prior to transfection, plasmid DNA was isolated using standard miniprep buffers (Omega Biotek) and silica membrane columns (Biobasic). To reduce batch-to-batch variability between plasmid DNA isolations and its potential impact on transfection efficiency, multiple plasmid DNA extractions were conducted in parallel, diluted to 50 ng/μL and pooled together.

Twenty-four hours prior to transfection, HEK293T cells were seeded at 65-75% confluency in 24-well plates. Prior to plating, wells were coated for 30 min with a 1 mg/mL solution of poly-D-lysine in PBS (MP Biomedicals Inc.) and washed with PBS (Gibco) once prior to media addition. Twenty-four to thirty hours after plating, 500 ng of SARS-CoV, SARS-CoV-2, or MERS 3CL$^{pro}$ expression plasmid was incubated with Opti-MEM and Lipofectamine 2000 for 30 min at room temperature prior to dribbling on cells, per manufacturer's protocol. In order to identify compounds with activity, 20 h after transfection, cells were washed once with PBS prior to adding 200 μL Trypsin-EDTA 0.25% (Gibco) in order to release cells from the plate. Trypsinized cell slurry was well mixed to ensure a single cell suspension. 96-well plates were coated with poly-D-lysine solution for 30 min and washed with PBS. Wells were filled with 100 μL of media ± drug and 1 μg/mL puromycin and were seeded with 9 μL of trypsinized cell slurry. After seeding into wells containing compound and puromycin, cells were incubated for 48 h.

The crystal violet staining protocol was adapted from previously published protocol[53]: After compound incubation with 3CL$^{pro}$ expressing cells in 96-well plates, the medium was discarded and cells were washed once with PBS. Cells were incubated with 50 μL of crystal violet staining solution (0.5% w/v crystal violet in 80/20 water/methanol solution) and rocked gently for 30 min. The staining solution was pipetted away and cells were washed four times with water using a multichannel pipette. Stained cells were left to dry for ≥4 h on the laboratory bench. The crystal violet staining solution was eluted upon the addition of 200 μL of methanol and gentle rocking over the course of 30 min. During rocking, plates were

sealed with parafilm to mitigate methanol evaporation. 100 µL of eluted stain from each well was transferred to a new 96-well plate for reading in a Tecan Infinite F50 plate reader with the software of Tecan iControl. The absorbance of each well was measured at 595 nm twice and values were averaged between replicate measurements. Blank wells were included in each batch of experiments, and absorbance values were normalized by background levels of staining from blank wells. For analysis of crystal violet staining experiments, relative growth was calculated from background normalized absorbance values.

**Fragment library screening.** A library consisting of 7247 diverse electrophile fragments (Enamine) was screened for inhibitory activity against the SARS-CoV-2 3CL protease. Screening was conducted by plating the compounds into 384 well plates, addition of the protease, addition of the substrate (MCA-AVLQSGFR-Lys(DNP)-Lys-NH2), and then incubating for 1 h at 30 ℃ before measuring fluorescence. The final concentrations used were 0.8 µM for the protease and 10 µM for the substrate in 5 µL assay volume. The assay buffer was 50 mM Tris-HCl, pH 7.5, 1 mM EDTA, 0.01% Triton X-100. On all plates, negative control wells of DMSO and positive control wells containing GC376 were included.

For each compound in the primary screening, the normalized raw percent inhibition was calculated based on raw values of DMSO wells (0% inhibition) and GC376 wells (100% inhibition). In addition, a smoothed score was generated by applying the b score method[54]. Hits were defined as compounds with values 2.5 standard deviations away from the mean in either the normalized raw percent inhibition or the smoothed score.

Compounds were initially tested at a concentration of 50 µM without replicates. There were 56 compounds that scored as hits from this screen, and all of these compounds were retested in triplicate at 50 µM again. All of the compounds except one were validated. The resulting 55 hits were then tested at 10 µM, 15 µM, 20 µM, 50 µM, 100 µM, and 200 µM in triplicate to generate dose-response curves and to calculate $IC_{50}$.

**Measurement of SARS-CoV-2 viral inhibition in Vero-E6 cells.** Stocks of SARS-CoV-2 strain 2019-nCoV/USA_WA1/2020 were propagated and titered in Vero-E6 cells. One day prior to the experiment, Vero-E6 cells were seeded at 30,000 cells/well in 96 well plates. Serial dilutions of the test compound were prepared in cell media (EMEM + 10% FCS + penicillin/streptomycin), overlaid onto cells, and then virus was added to each well at an MOI of 0.2. Cells were incubated at 37 ℃ under 5% CO2 for 72 h and then the supernatant was collected from all wells. Viral RNA was extracted using PureLink™ Pro 96 Viral RNA/DNA Purification Kit (Invitrogen) according to the manufacturer's instructions, and then quantified against a RNA standard by quantitative reverse-transcriptase PCR (qRT-PCR) using TaqPath™ 1-Step RT-qPCR Master Mix, CG (Applied Biosystems) with primers 5′-GACCCCAAAATCAGCGAAAT-3′ and 5′-TCTGGTTACTGCCAGTTGAAT CTG-3′ and probe 5′-FAM-ACCCCGCATTACGTTTGGTGGACC-BHQ1-3′.

Alternatively, also at 72 h post-infection, viral cytopathic effect was scored in a blinded manner. Inhibition was then calculated by comparison to control wells with no inhibitor added.

EC50 values were determined by fitting an asymmetric sigmoidal curve to the data (GraphPad Prism). Cells were confirmed as mycoplasma negative prior to use. All experiments were conducted in a biosafety level 3 (BSL-3) lab.

**Measurement of SARS-CoV-2 viral inhibition in Huh-7^ACE2 and Huh-7^ACE2+TMPRSS2 cells.** Inhibition of SARS-CoV-2 by compounds was tested in Huh7 cells (human hepatocellular carcinoma, gift of Matthew Evans, Mount Sinai) overexpressing human ACE2 (Huh-7) or human ACE2 and human TMPRSS2 (Huh-7^ACE2+TMPRSS2). Huh-7^ACE2 cells were prepared by transducing Huh7 cells with lentivirus encoding human ACE2, packaged using pLEX307-ACE2-blast (Addgene plasmid #158449), pMD2.G (Addgene plasmid #12259, gift of Didier Trono), and psPAX2 (Addgene plasmid #12260, gift of Didier Trono), and selecting for stable expression with 5 µg/mL blasticidin. Huh-7^ACE2+TMPRSS2 cells were prepared by transducing Huh-7^ACE2 cells with lentivirus encoding human TMPRSS2, packaged using pLEX307-TMPRSS2-puro (Addgene plasmid #158457), pMD2.G, and psPAX2, and selecting for stable expression with 2 µg/mL puromycin.

One day prior to the experiment, 96-well clear bottom black plates (Corning) were treated with poly-D-lysine (Gibco) according to the manufacturer's instructions, and then 20,000 Huh-7^ACE2 or Huh-7^ACE2+TMPRSS2 cells were seeded in each well in DMEM + 10% FCS + penicillin/streptomycin.

The test compound was then serially diluted in DMEM + 10% FCS + penicillin/streptomycin, and added as appropriate to wells. At the same time, 0.05 MOI of SARS-CoV-2-mNeonGreen (a fluorescent reporter variant of strain 2019-nCoV/USA_WA1/2020, gift of Pei-Yong Shi) was added to wells[55]. Control wells without treatment and with virus only were included on all plates.

Twenty-four hours post-infection, the supernatant was removed and all wells were fixed with 4% PFA in PBS (Fisher Scientific) containing 2 µg/mL DAPI (Fisher Scientific). The supernatant was removed after 45 min incubation at RT, and wells were washed with sterile PBS, before being imaged on an IN Cell Analyzer 2000 (GE Healthcare). Cells were then scored for infection using CellProfiler 3.0[56] or Plaque2.0[57].

All experiments were conducted in a biosafety level 3 (BSL-3) lab. Cells were confirmed mycoplasma negative prior to use.

**Measurement of SARS-CoV-2 viral inhibition in Caco-2 cells.** This measurement was performed at the Institute for Antiviral Research of Utah State University under a contract sponsored by NIAID. Active compounds are further tested in a confirmatory assay. Confluent or near-confluent cell culture monolayers of Caco-2 cells are prepared in 96-well disposable microplates the day before testing. Cells are maintained in MEM supplemented with 5% FBS. The test compound is prepared at a serial dilution of concentrations. Three microwells are used per dilution. Controls for the experiment consist of six microwells that are infected and not treated (virus controls) and six that are untreated and uninfected (cell controls) on every plate. A known active drug is tested in parallel as a positive control drug using the same method as is applied for test compounds. The positive control is tested with every test run.

Growth media is removed from the cells and the test compound is applied in 0.1 ml volume to wells at 2X concentration. Virus, normally at ~60 CCID$_{50}$ (50% cell culture infectious dose) in 0.1 ml volume is added to the wells designated for virus infection. Medium devoid of virus is placed in cell control wells. Plates are incubated at 37 ℃ with 5% CO$_2$. After sufficient virus replication occurs (3 days for SARS-CoV-2), a sample of supernatant is taken from each infected well (three replicate wells are pooled) and tested immediately for reduction of virus yield (VYR) or held frozen at −80 ℃ for later virus titer determination.

The VYR test is a direct determination of how much the test compound inhibits virus replication. Virus yielded in the presence of test compound is titrated and compared to virus titers from the untreated virus controls. Titration of the viral samples (collected as described in the paragraph above) is performed by endpoint dilution. Serial 1/10 dilutions of virus are made and plated into 4 replicate wells containing fresh cell monolayers of Vero 76 cells. Plates are then incubated, and cells are scored for presence or absence of virus after distinct CPE is observed, and the CCID$_{50}$ calculated using the Reed–Muench method[58]. The 90% effective concentration (EC$_{90}$) is calculated by regression analysis by plotting the log$_{10}$ of the inhibitor concentration versus log$_{10}$ of virus produced at each concentration. EC$_{90}$ values were calculated from data to compare to the concentration of drug compounds as measured in the pharmacokinetic experiments. Drug concentrations in critical tissues above EC$_{90}$ values are targeted as for clinical relevant applications (instead of EC$_{50}$ values).

**Measurement of SARS-CoV, MERS-CoV, HCoV-229E, HCoV-OC43, and Influenza H1N1 viral inhibition in cells.** These measurements were performed at the Institute for Antiviral Research of Utah State University under contracts sponsored by NIAID. SARS coronavirus (strain: Urbani) was tested in Vero 76 cells, with M128533 being used as control drug at 0.1–100 µg/ml. MERS Coronavirus (strain: EMC) was tested in Vero 76 cells, with M128533 being used as control drug at concentration of 0.1–100 µg/ml. Human coronavirus alpha (strain: 229E) was tested in Huh-7 cells, with M128533 being used as control drug at concentration of 0.01–10 µg/ml. Human coronavirus beta (strain: OC43) was tested in RD cells, with M128533 being used as control drug at concentration of 0.01-10 µg/ml. Influenza A H1N1 (strain: California/07/2009) was tested in MDCK cells, with Ribavirin being used as control drug at concentration of 1–1000 µg/ml. EB46, EB54, and NK01-63 were dissolved in DMSO and tested in the concentration range of 0.1–100 µM, with DMSO being used as vehicle control. Visual inspection and neutral red assay were used to measure cytopathic effect and toxicity. EC$_{50}$ and CC$_{50}$ values were extracted from primary data. EC$_{50}$ and CC$_{50}$ results from visual inspection and neutral red assay were consistent, while neutral red assay data were reported here in the paper as representatives. EC$_{50}$ of control drugs on SARS-CoV, MERS-CoV, HCoV-229E, HCoV-OC43, and Influenza H1N1 are <0.1 µM, 0.16 µM, 0.021 µM, 0.018 µM, and 3.2 µM, respectively.

**Measurement of cytotoxicity.** Huh7^ACE2 cells were plated at 1,000 cells per well in white 384-well plates (36 µL per well) in triplicate and incubated overnight. The cells were then treated with 4 µL medium containing a two-fold dilution series of vehicle (DMSO) or 3CL protease inhibitors (starting from 50 µM). After 48 h incubation, 40 µL of 50% CellTiter-Glo (Promega) 50% cell culture medium was added to each well and incubated at room temperature with shaking for 10 min. Luminescence was measured using a Victor X5 plate reader (PerkinElmer). All cell viability data were normalized to the DMSO vehicle condition. From these data, dose-response curves and CC$_{50}$ values were computed using Prism software (GraphPad).

**Crystallization, data collection, and structure determination.** To generate the complex of SARS-CoV-2 3CL protease bound to inhibitors, 50 µM of the 3CL protease was incubated with 500 µM of the inhibitors in a buffer comprised of 50 mM Tris-HCl (pH 7.5), 1 mM EDTA for 1 h at 4 ℃. The complex was then concentrated to 5 mg/mL using a 10 kDa concentrator. Crystals of 3CL in complex with GC376, which were initially obtained from the High-Throughput Crystallization Screening Center of the Hauptman-Woodward Medical Research Institute (HWI) (https://hwi.buffalo.edu/high-throughput-crystallization-center/) were

reproduced using under oil micro batch method at 4 °C[37,59]. These crystals were subsequently used as seeds for growing crystals of all 7 complexes reported here.

Block-like crystals of 3CL in complex with inhibitors EB46, EB48, EB54, EB56, SL-4-241, NK01-14, NK01-48, and NK01-63 were grown using a crystallization reagent comprising 0.1 M potassium thiocyanate, 0.1 M sodium acetate (pH 5), and 20% (w/v) PEG 8000 with protein to crystallization reagent ratio of 2(or 3):1 µL. All crystals were subsequently transferred into a similar crystallization reagent that was supplemented by 20% (v/v) ethylene glycol and flash frozen in liquid nitrogen. A native dataset was collected on each crystal of the protein-inhibitor complex at the NE-CAT24-ID-C beam line of Advanced Photon Source in Lemont, IL. Crystals of 3CL in complex with EB46, EB48, EB54, EB56, SL-4-241, NK01-14, NK01-48, and NK01-63 diffracted the X-ray beam to resolution 1.65 Å, 2.08 Å, 1.68 Å, 2.03 Å, and 2.17 Å, 1.64 Å, 1.79 Å, 1.55 Å, respectively. The images were processed and scaled in space group C2 using XDS[60]. The structure of each protein-inhibitor complex was determined by molecular replacement method using MOLREP[61] program and crystal structure of 3CL in complex with GC376 (PDB id: 7JSU) as a search model. The geometry of each crystal structure was subsequently fixed and the corresponding inhibitor was modeled by program XtalView[62] and COOT[63], and refined by Phenix[64] program. Figures of crystal structures, including inhibitor electron density plots, were prepared with the software of Pymol. There is one protomer of 3CL complex in the asymmetric unit of crystal. The crystallographic statistics is shown in Supplementary Tables 4 and 5.

**Analysis of low energy solution conformation of small molecules**. Small molecules were imported into Maestro of Schrodinger Suite and Desmond molecular dynamics calculations were performed to generate a low energy solution conformation of the small molecules.

**Mouse maintenance**. All animal procedures were approved by the Columbia University Institutional Animal Care and Use Committee (IACUC) under protocol number AC-AAAZ4461, and were performed in accordance with all relevant regulatory standards. This study involves 88 C57BL/6 mice ordered from Jackson Lab. 44 mice are male, while the other 44 are female. Mice were housed 4 per cage under 12 h light/dark cycle conditions. Ambient temperature is 22 °C and humidity is 50%. Food and water were provided ad libitum.

**In vivo safety study of NK01-63**. The following protocol for animal study (AC-AAAZ4461) was approved by Columbia IACUC. We performed a toxicity study to evaluate whether NK01-63 has any unexpected toxicity following chronic dosing regimens. Therefore, 40 healthy 8-week old C57BL/6 mice were randomly divided into four groups of 10 mice: (1) IP, 20 mg/kg NK01-63; (2) IP, water vehicle; (3) PO, 20 mg/kg NK01-63; (4) PO, water vehicle. Each group has 5 males and 5 females. NK01-63 powder was dissolved in water as 2 mg/ml solution. The mice were treated with 20 mg/kg NK01-63 or water vehicle via intraperitoneal (IP) or oral gavage (PO) dose once per day for 14 consecutive days. Body weight of each mouse was recorded every day. Body weights were then normalized to the initial body weight on the first day to analyze the effects of treatment throughout the 14-day treatment.

**In vivo pharmacokinetic study of NK01-63**. The following protocol for animal study (AC-AAAZ4461) was approved by Columbia IACUC (institutional animal care and use committee). For in vivo administration, NK01-63 powder was dissolved in water as 2 mg/ml solution. NK01-63 were delivered at 20 mg/kg by intraperitoneal (IP) injection or oral gavage (PO) to 8-week old C57BL/6 mice. After 0 h, 2 h, 4 h, 8 h, or 24 h, animals were decapitated to collect blood and dissect fresh lung tissue. For each time point of each route of administration (ROA), 4 mice (2 males and 2 females) were included in the study. After euthanasia by CO₂, blood was collected with cardiac puncture. Blood samples were treated with EDTA solution (50 mM final concentration) on ice to prevent coagulation and centrifuged at $2100 \times g$ for 10 min at 4 °C, with the plasma collected and placed into a clean Eppendorf tube, flash frozen, and stored at −80 °C prior to use. Lung tissue is collected immediately post sacrifice and flash frozen and stored at −80 °C prior to use.

To a 100 µL aliquot of plasma collected from particular time point and ROA, 900 µL of methanol (MS grade) was added and the sample was mixed by rocking for 5 min prior to sonicating. Extractions were performed overnight at 4 °C to ensure optimal extraction. After extractions, samples were rocked and centrifuged for 10 min at $4000 \times g$ at 4 °C. The supernatant was collected for the following LC-MS analysis.

A known amount of lung tissue collected from particular time point and ROA was placed into a bead lysing tube (Omni International). To this, an appropriate volume of water (MS grade) to result in 500 mg/mL mixture. The samples are homogenized using a bead homogenizer for three cycles at speed 5 for 30 seconds, with sample placed on ice between cycles. After homogenizing a 100 µL aliquot of the tissue homogenate was collected, and 900 µL of methanol (MS grade) is added. The samples were mixed by rocking for 5 min prior to sonicating. Extractions were performed overnight at 4 °C to ensure optimal extraction. After extractions, samples were rocked and centrifuged for 10 min at $4000 \times g$ at 4 °C. The supernatant was collected for the following LC-MS analysis.

UPLC-MS analysis was performed on Synapt G2-XS (Waters Corp., U.K) mass spectrometer, equipped with an electrospray ionization source. Chromatographic separation was performed by injecting 8 µL of the sample in duplicates onto an Acquity Premier BEH C18 column (1.7 µm, 2.1 × 50 mm) maintained at 50 °C, over a 3 min gradient. The flow rate was maintained at 0.5 mL/min. The initial flow conditions were 50% solvent A (water containing 0.1% formic acid) and 50% solvent B (methanol containing 0.1% formic acid) and the gradient was held for 0.25 min. Solvent B was raised to 100% by 1 min and returned to the initial condition by 2.00 min, and the column was equilibrated for additional 1.00 min. The data were acquired in positive ionization mode, over a mass range of $m/z$ 50–1000, using the following parameters—capillary voltage of 0.5 kV, sampling cone voltage of 32 V, source temperature of 120 °C, desolvation temperature of 400 °C and desolvation gas flow rate of 800 L/h. Leucine enkephalin ($m/z$ 556.2771 $[M+H]^+$) was used as a lock mass via lockspray interface for mass accuracy. The compound concentration in each plasma or lung tissue sample was determined using QuantLynx (Waters Corp.) with a NK01-63 standard calibration curve. In particular, the base peak chromatogram at $m/z$ 472.2 (corresponding to $[M+H]^+$ of the aldehyde (active) form of NK01-63) with a width of ±0.1 was integrated and peak area quantified.

Pharmacokinetic data were assessed using GraphPad Prism with lognormal of one-phase exponential decay.

**Statistical information**. The data reported in this study represent the mean and s.d. of at least three independent experiments measured in triplicate, unless otherwise stated in the figure legend. For statistical analyses, two-tailed $t$ test was performed to assess whether a significant difference exists between two groups of samples. A $P$ value of less than 0.05 was considered statistically significant. All statistical tests were carried out using Prism 8 software (GraphPad).

**Reporting summary**. Further information on research design is available in the Nature Research Reporting Summary linked to this article.

## Data availability

The data that support this study are available from the corresponding author upon reasonable request. Structural data for the SARS-CoV-2 3CL protease in complex with EB46, EB48, EB54, EB56, SL-4-241, NK01-14, NK01-48, and NK01-63 were deposited in the Protein Data Bank (PDB), with accession codes 7TIU, 7TIV, 7TIW, 7TIX, 7TJ0, 7TIA, 7TIY, and 7TIZ. All other data generated in this study are provided in the Supplementary Information and Source Data file. Publicly available dataset used in this study is the crystal structure of SARS-CoV-2 3CL protease in complex with GC376, with accession code PDB ID: 7JSU. Source data are provided with this paper.

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

## Acknowledgements

This work was supported by a grant from the Jack Ma Foundation to D.D.H. and A.C. and by grants from Columbia Technology Ventures and the Columbia Translational Therapeutics (TRx) program to B.R.S. A.C. is also supported by a Career Awards for Medical Scientists from the Burroughs Wellcome Fund. S.I. is supported by NIH grant T32AI106711. E.B. thanks the Experientia Foundation for a postdoctoral fellowship. We thank the staff of the High-Throughput Crystallization Screening Center of the Hauptman-Woodward Medical Research Institute for screening crystallization conditions. Crystallization screening was supported through NSF grant 2029943. This research used resources of the Advanced Photon Source, a U.S. Department of Energy (DOE) Office of Science User Facility, operated for the DOE Office of Science by Argonne National Laboratory under Contract No. DE-AC02-06CH11357. Extraordinary facility operations were supported in part by the DOE Office of Science through the National Virtual Biotechnology Laboratory, a consortium of DOE national laboratories focused on the response to COVID-19, with funding provided by the Coronavirus CARES Act. Funding for experiments completed at Utah State University was provided by the Respiratory Diseases Branch, National Institute for Allergy and Infectious Diseases, NIH USA (Contract N01-AI-30048). M.F. thanks the Coordenação de Aperfeiçoamento de Pessoal de Nível Superior-Brazil (CAPES/PRINT-UFSC, Finance Code 001) for a Visiting Research scholarship.

## Author contributions

B.R.S., D.D.H., and A.C. conceived and implemented the project. H.L., T.R., A.C., D.D.H., and B.R.S. planned and designed the experiments. S.I. and S.J.H. cloned, expressed, and purified proteins. C.Q. performed isothermal titration calorimetry. H.L. conducted the protease inhibition assay. S.I. and H.M. performed the qRT-PCR viral assay. H.L. performed the human protease selectivity assay. H.L., B.F., W.L., H.S., C.J., M.A.L., and T.M. performed mass spec analysis of protein-inhibitor complex. M.E.S. and B.F. determined the microsomal and plasma stability of lead compounds. B.F. identified the metabolites of compound 4 with LC-MS/MS. S.J.R. performed the HEK293T-based protease inhibitor assay. S.I. and C.K. conducted the fragment library screen. S.I., M.S.N., and Y.H. performed the CPE-based viral assay. S.I. performed the Huh7[ACE2]-based antiviral assay. B.H. performed Caco-2-based SARS-CoV-2 antiviral assay and cellular antiviral assay against other coronaviruses. H.L. and F.F. crystallized the proteins. F.F. collected diffraction data and solved the crystal structures. F.F and H.L. analyzed the structural data. A.Z. designed and synthesized compound 4, sulfonamide analogs, compound 4 analogs, and contributed to the design and synthesis of GC376 analogs. N.K., E.B., and S.L. designed and synthesized GC376 analogs. N.E.S.T. designed and synthesized sufonylpyridine analogs. H.L., X.X., J.D.D., M.B., M.F., and P.R. performed in vivo study. H.L. and B.R.S. wrote the manuscript with input from all authors.

## Competing interests

S.I., H.L., A.Z., B.R.S., A.C., T.R., N.K., E.B., F.F., N.E.S.T., S.L., and D.D.H. are inventors on invention disclosures and patent applications submitted based on this work. B.R.S. is an inventor on additional patents and patent applications related to small molecule therapeutics, and co-founded and serves as a consultant to Inzen Therapeutics, Nevrox Limited, Exarta Therapeutics, and ProJenX Inc., and serves as a consultant to Weatherwax Biotechnologies and Akin Gump Strauss Hauer & Feld LLP. A.Z. is an inventor on additional patents and patent applications related to small molecule therapeutics, and co-founded and serves as a consultant to ProJenX Inc. C.Q., W.L., H.S., C.J., M.A.L., and T.M. are employed by Waters Corporation. The remaining authors declare no competing interests.
