## [Peer Review File · Nature Communications]

Development of optimized drug-like small molecule inhibitors of the SARS-CoV-2 3CL protease for treatment of COVID-19Reviewers' Comments:

Reviewer #1:

Remarks to the Author:

OVERALL ANALYSIS

Liu and coworkers have written a comprehensive article on the development of drug-like small molecule inhibitors of the SARS-CoV-2 3CL protease. The topic is highly relevant in the current Corona pandemic, since compounds for the treatment of the Covid-19 disease may arise from the obtained results.

Undoubtedly, the study is highly significant in the field and among the most advanced ones compared to recent similar studies, which are properly cited in the reference section. There is even a potential to impact pharmacology and medicine in the near future. The most noteworthy result is that at least one of the optimized inhibitors has a high potential as lead compound. NK01-63, termed coronastat shows low nM inhibition of the 3CL protease and is highly efficient in blocking SARS-CoV-2 infection of various cultured cells.

Regarding the claims and conclusions, I see more than sufficient evidence in the data and results. While I really appreciate the multidisciplinary approach with many interesting and time consuming experiments, I have to state that the enormous amount of data is not very well presented and, thus, confusing. Although the article is well written and clear in most parts, it is extremely lengthy with the extended data section, which is quite inaccessible for most readers, except for a few specialists.

The quality of the data is overall technically sound and they are obtained with appropriate methods. However, there are some notable exceptions in the supplementary section, e.g. in Extended Data Figures 1 and 2. The dose response plots seem to be inappropriate and should be replaced by standard inhibition curves. Details can be found under the Minor Points. In general, the data analyses and interpretations meet the expected standards, but again with some exceptions. Firstly, data for IC50 calculation should be fitted with logarithmic curves, not with asymmetric sigmoidal ones. Secondly, the calculation of Ki values for covalent inhibitors appears to be a serious flaw. Sometimes only duplicates were measured, which is not standard in enzyme kinetics. These experiments should be repeated by measuring triplicates for a revised version. The Ki values have to be replaced by kinact/KI values in a revised version.

Mostly, the methodology is sound, while the enzyme kinetics has several parts, which can be largely improved. I have this already mentioned for the corresponding data analyses, which is currently the weakest part of the study. Therefore, I see no problems for the validity of the overall approach, which comprises enzyme kinetics, calorimetry, viral infection assays in cell cultures, crystal structure determination, and further experiments. In addition, the authors provide sufficient details in the methods to reproduce each part of the study.

In the sections above, I suggested several improvements for the study. Eventually, the authors should delete the comparison of the 3CL protease with Caspase-3 and replace it with Cathepsin L and/or B. Another critical point is the inconsistency of the dose response values EC50 and EC90 without explanation. Also, the usage of the various cell cultures (Vero-E6, 293T, Huh-7ACE, Caco-2, etc.) is confusing and should be explained. The Discussion is simply a summary with an outlook, which has to be newly written in an appropriate way. However, the most crucial point is a proper presentation of the data in the figures. The majority of them is far too small, in particular, most labels are close to invisibility. My final assessment is a major revision. Otherwise, the study has great potential and could become a real groundbreaking publication.

MINOR POINTS

Page 1, 7-8: There is no reference to the "*", e.g. for authors Rovis to Stockwell.

P2, 43-44: I guess such abbreviations should be avoided in the abstract, at least explained upon first usage.

44-45: "the first small molecule protease inhibitors" singular/plural error.

P6, 110-111: The mismatch and inconsistency of the ITC values and IC50 is striking. It should be mentioned in the Discussion. Why was Ebselen not measured?

119-120: "an IC50" values. Seemingly, some sections were written in non-standard English.

P7, 133: Structural formula of GC376 should be shown in a corresponding figure.

P8, 160-161: "as predicted based on chemical logic" is not such a convincing argument. I would prefer can be expected or something similar.

165-169: The sentence is confusing and too long. Thus, It should be split and reworded.

P9, 179: Caspase-3 is only active as dimer, its specificity is Asp in P1. A comparison with the 3CL protease makes no sense and, thus, caspase-3 should be deleted from the article.

182: Also, the specificity of the serine protease chymotrypsin is quite different compared to 3CL, but it is acceptable.

P10,194: It is not clear why EC90 values were used, and not EC50. A comparison with the IC50 might be easier, otherwise explain it.

P13, 247: The abbreviation CYP has to be explained.

265: "vulnerable to metabolism" sounds a bit weird, do reword it.

P14, 270: "didn't" is colloquial, it has to be changed to "did not".

P15, 295: Ki values for covalent inhibitors make no sense. Use kinact/KI (e.g. Krippendorff et al, 2009, J. Biomolecular Screening)

299: Why now EC50, when before EC90 was used? Consistency is required.

P16, 316: Now it is EC90 again. Use nM values.

P17, 327: Not "1 equivalent", but one equivalent; analogs.

P17, 331: I wonder, whether it is possible to obtain a true Kd for covalent inhibitors from ITC. I doubt it seriously and think it must be a counterpart to kinact/KI.

P18, 357/367: did not

P19, 378-381: The sentence is confusing, rewrite it.

391: "Strong no-polar"? How is the strength measured? Is it just a hydrophobic interaction, which is usually not that strong.

P20, 394-396: Sentence requires rewording.

398-399: "covalently but reversibly" seems contradictory, while aldehydes may be over time hydrolyzed. Anyway, some proof is needed, not simple conjectures.

P21-22, 435-439: The long sentence is not very clear.

P22, 444: Why is not EC50, use nM values.

448: within the cellular context.

451: coronaviruses

P23, 464-465: EC90 again, should 100 nM; with an EC50 value.

476: IND is not explained.

P24-25: The Discussion is a mere summary with an outlook. This section requires extensive rewriting.

P43, 872/878: K_d for covalent inhibitors appears to be inappropriate.

874/876: I guess "heat" is not the proper expression, so I assume it is the free energy.

P44, 895: Standard for IC_{50} data is fitting $\log(\text{inhibitor})$ vs normalized response. Asymmetric sigmoidal curves might completely distort the true values. I suggest to recalculate all IC_{50} data accordingly.

906: The standard for enzyme kinetics is triplicate data, so repeat the experiment accordingly.

910: K_i values for covalent inhibitors cannot be calculated properly, use k_{inact}/K_I values.

P62, 1314: Nowadays, it is standard to deposit structural data in the Protein Data Bank (PDB) upon submission of the manuscript. Thus, the final PDB codes have to be included in the manuscript. Of course, the corresponding PDB data should have the status "hold for publication" and are only released, when the paper is accepted/published.

P66-69: The figures 1, 2, 3 and 5 are far too small, which applies to the molecule structures and especially to all the tiny labels. New, larger versions have to be prepared.

P68, 1394: I am not familiar with the standard procedures of ITC, but duplicates seem dubious to me. I would prefer triplicates.

1402-1407: eq should be written out at least for the first occurrence.

P69, 1423: What is MOI?

1427: mouse – which cell type?

P70: Table 1. Display IC_{50} values, not K_i . The resolution of the crystal structure does not contribute any information on inhibition. Delete the column.

P71: The labels for the residues have to be enlarged.

P74-75: Extended data Figures 1 and 2. As in previous figures the labels are too small. The molecular structures are nearly invisible, data points and curve quite faint. Why are they shown as dose response curves? Show the plots as is standard. I guess that sigmoidal plots were applied not standard $\log(\text{conc inhibitor})$ ones. Plots 2, 4, 8, 17, 19, 20, 21, 31, 32, 36, 38, 39, 42, 43, 44, 49, 52, 53, 54, and 56 make not much sense. Either replot them as log curves, or remeasure them. Otherwise delete them. It is impossible to calculate proper IC_{50} with this approach, when so many curves start at 25% or even 50% inhibition.

P76-78: Extended data Figures 3, 4, and 5. Labels are too small. Figure 5D: A comparison with chymotrypsin might be acceptable, because of the fold. However, elastase would be the better choice as the specificity is more similar. Caspase-3 is a bad choice and should be replaced by Cathepsin B or preferentially Cathepsin L.

P80: Extended data Figure 6. In panels A, B and C the labels are just ok, while in D, E and F the are too small again. In 6D K_i values are displayed, which is basically not possible for covalent inhibitors, calculate k_{inact}/K_I .

P81, 1602: Duplicates are problematic, perhaps acceptable for ITC.

P82: Extended data Figure 7. All panels require general enlargement.

P84: Extended data Figure 8. Panel A has to be enlarged, while B and C are ok.

P87-88: Extended data Figure 9: Except for panels B and C significant enlargement is required. 1723-1724: Ki problem and only duplicates; 1743: The duplicate in ITC might be ok.

P90: Extended data Figure 11. 2Fo-Fc and Fo-Fc maps cannot be distinguished for all panels. The 3s contour is unusual for the 2Fo-Fc map (typically 1s). The figures would be more convincing, if 2Fo-Fc omit maps were shown, calculated without the inhibitor model.

P92-93, 1816-1817: Extended data Figure 12. Correct to "and the respective 3CL protease in green"

P94-95: Extended data Figure 13. Labels in panels B and D are just a bit to small, the rest is ok.
1852-1854: The Ki problem and only duplicates.

P96: Extended data Table 1. It is alright to analyze for all these proteases with respect to interference of the compounds. Obviously, Cathepsins B and especially L are affected by the compounds EB54 and NK01-63. They are cysteine proteases and their specificity is closer to the 3CL protease than chymotrypsin or caspase-3 according to the MEROPS database. 1867: Better show the IC50 value for the 3CL protease.

P98: Extended data Table 3. Z-FR-AMC is certainly not the best substrate for Cathepsin B and L. In case of CatB Gly and Ala are preferred in P1 position, CatL prefers Gly, Ala, Ser, and Thr.

Reviewer #2:

Remarks to the Author:

Dear authors:

Liu et al designed a series of SARS-CoV-2 inhibitors based on the previous coronavirus 3CL protease inhibitor GC376. Binding assay, cellular antiviral assay and co-crystallization all showed they have promising results. Furthermore, these compounds have pan-coronavirus activity. Among them, NK01-63 is promising.

Overall, this study is technically sound, the results are interesting in the field of the drug discovery for COVID-19, and the manuscript well written. Development of an effect antiviral is very important to control the pandemic. The limitation is lack of in vivo study, including the stability, toxicity and metabolism of the compounds in animal, efficacy of treatment on infected animals.

Specific questions:

1. Since they are free-cystein reactive, I suggest authors to test the inhibition against other important human cystein proteases such as caspase-3, cathepsin K, S etc.
2. In vivo toxicity is still of my concern. If possible, I suggest authors at lease evaluate the short term toxicity in animals.
3. I'd suggest authors to compare with other first line antivirals such as Remdesivir in invo and in vitro.

Reviewer #3:

Remarks to the Author:

This manuscript describes extensive studies on analogs of GC376 and other small molecules as inhibitors of the main protease (3CL) of the SARS2 virus that causes COVID19. The work is well done, and the authors have found several potent inhibitors that show promising properties as drugs, including the compound they call coronastat.

The main difficulty with this manuscript is lack of novelty.

After the initial publication of Hilgenfeld and those of other groups in 2020 (their refs 1-10 in the submitted paper, a number of analogs of GC376 were examined for enzyme inhibition, by crystallography of enzyme-inhibitor complexes, and for antiviral activity in cells (e.g. ref 26 in their paper). Some of these analogs, including from ref 26, are re-reported without citation in the present paper as new experimental procedures and results. This is not acceptable. This reviewer doubts that the incremental advances in the present paper merit publication in this journal. The authors use the word "we" very extensively in their paper, often disguising the fact that much has been done before and they are simply repeating previous publications.

There is another problem. Their claim that "...GC376 was soluble in PBS buffer at all concentrations tested (up to 1 mM, Fig. 3c.." is not agreement with other studies. That is why it is injected in aqueous ethanol solution into cats and other animals in the literature. It does form clear micelles and colloid suspensions when concentrated in water (see ref 26 in their paper) but such administration to humans requires additional testing in animals. Injection of aqueous ethanol solutions or DMSO solutions into humans is not feasible.

Revision of manuscript NCOMMS-21-31169

The following includes a **point-by-point response to the reviewer comments** and our corresponding revisions.

Reviewer #1 (Remarks to the Author):

OVERALL ANALYSIS

Liu and coworkers have written a comprehensive article on the development of drug-like small molecule inhibitors of the SARS-CoV-2 3CL protease. The topic is highly relevant in the current Corona pandemic, since compounds for the treatment of the Covid-19 disease may arise from the obtained results.

Undoubtedly, the study is highly significant in the field and among the most advanced ones compared to recent similar studies, which are properly cited in the reference section. There is even a potential to impact pharmacology and medicine in the near future. The most noteworthy result is that at least one of the optimized inhibitors has a high potential as lead compound. NK01-63, termed coronastat shows low nM inhibition of the 3CL protease and is highly efficient in blocking SARS-CoV-2 infection of various cultured cells.

Regarding the claims and conclusions, I see more than sufficient evidence in the data and results. While I really appreciate the multidisciplinary approach with many interesting and time consuming experiments, I have to state that the enormous amount of data is not very well presented and, thus, confusing. Although the article is well written and clear in most parts, it is extremely lengthy with the extended data section, which is quite inaccessible for most readers, except for a few specialists.

Response: We thank the reviewer for the positive comments. As suggested, we have streamlined our manuscript during revision to cut down the length and present concisely.

The quality of the data is overall technically sound and they are obtained with appropriate methods. However, there are some notable exceptions in the supplementary section, e.g. in Extended Data Figures 1 and 2. The dose response plots seem to be inappropriate and should be replaced by standard inhibition curves. Details can be found under the Minor Points. In general, the data analyses and interpretations meet the expected standards, but again with some exceptions. Firstly, data for IC₅₀ calculation should be fitted with logarithmic curves, not with asymmetric sigmoidal ones. Secondly, the calculation of K_i values for covalent inhibitors appears to be a serious flaw. Sometimes only duplicates were measured, which is not standard in enzyme kinetics. These experiments should be repeated by measuring triplicates for a revised version. The K_i values have to be replaced by k_{inact}/K_i values in a revised version.

Response: We did fit the IC₅₀ data with Log(inhibitor) vs. normalized response curve

(the standard inhibition curve), but we described the method incorrectly. We thank the reviewer for catching this. We have corrected our description of the methods in the Method section. But still, as the reviewer suggested, we have checked every IC50 curve for the correctness of curve being used.

As the reviewer pointed out, inhibition constants (K_i) are not suitable for evaluations of covalent inhibitors, as it works better for non-covalent ligands. Therefore, following the reviewer's suggestion, instead of K_i , we now use k_{inact}/K_i values to characterize and compare our most potent covalent inhibitors in the revised manuscript. We also included explanation of this application of k_{inact}/K_i values in the Discussion section.

Mostly, the methodology is sound, while the enzyme kinetics has several parts, which can be largely improved. I have this already mentioned for the corresponding data analyses, which is currently the weakest part of the study. Therefore, I see no problems for the validity of the overall approach, which comprises enzyme kinetics, calorimetry, viral infection assays in cell cultures, crystal structure determination, and further experiments. In addition, the authors provide sufficient details in the methods to reproduce each part of the study.

In the sections above, I suggested several improvements for the study. Eventually, the authors should delete the comparison of the 3CL protease with Caspase-3 and replace it with Cathepsin L and/or B. Another critical point is the inconsistency of the dose response values EC50 and EC90 without explanation. Also, the usage of the various cell cultures (Vero-E6, 293T, Huh-7ACE, Caco-2, etc.) is confusing and should be explained. The Discussion is simply a summary with an outlook, which has to be newly written in an appropriate way. However, the most crucial point is a proper presentation of the data in the figures. The majority of them is far too small, in particular, most labels are close to invisibility. My final assessment is a major revision. Otherwise, the study has great potential and could become a real groundbreaking publication.

Response:

1. As suggested, we have removed the comparison of 3CL protease with Caspase-3, due to the lack of similarity in substrate specificity between 3CL and Caspase-3. Instead, we tested our compounds against Cathepsin L and B and reported the results accordingly.
2. We agree with the reviewer that consistently reporting EC50 values throughout the manuscript instead of EC90 values would work better, especially considering the comparison with IC50 values. We have therefore revised the manuscript to replace all EC90 values by EC50 values whenever applicable.
3. We thank the reviewer for suggesting to discuss and explain the usage of the various cell culture systems. We now provided explanation of the usage of the various cell lines in the discussion.
4. We have adjusted the appearance of all related figures for better visualization.

Overall, we really appreciate the in-depth and considerate comments and suggestions provided by the reviewer, who had a deep understanding of our work and helped us improve all aspects of the manuscript, from experiment design to data presentation, from overall outline to words and grammar.

MINOR POINTS

Page 1, 7-8: There is no reference to the “*”, e.g. for authors Rovis to Stockwell.

Response: We thank the reviewer for catching this. “*” refers to corresponding authors. Reference to “*” as label of corresponding author is now added for clarifications in the revised manuscript.

P2, 43-44: I guess such abbreviations should be avoided in the abstract, at least explained upon first usage.

Response: We thank the reviewer for reminding us to avoid using abbreviations without explanation upon first usage. We have checked all abbreviations in the manuscript and made sure we explained each of the abbreviations upon first usage in the revised usage. In this particular case, we replaced abbreviations.

44-45: “the first small molecule protease inhibitors” singular/plural error.

Response: We thank the reviewer for catching this singular/plural error and other typos or grammar mistakes in the manuscript. We have accordingly corrected all typos or grammar mistakes in the revised manuscript as mentioned here and below.

P6, 110-111: The mismatch and inconsistency of the ITC values and IC50 is striking. It should be mentioned in the Discussion. Why was Ebselen not measured?

Response:

1. We agree with the reviewer that the biophysical K_d values appeared not to be matching with the biochemical IC_{50} values, potentially due to the covalent natures of the compounds being tested. As the reviewer pointed in the comments below, disassociation constants (K_d) are not ideal for evaluations of covalent inhibitors, as it can only provide the apparent binding affinities for covalent ligands and works better for non-covalent ligands. Instead, following the reviewer’s suggestion, we used k_{inact}/K_i to characterize and compare our most potent covalent inhibitors in the revised manuscript. For ITC data, we now only reported stoichiometry and binding energy, but not “ K_d ”, in the revised manuscript. For this particular part, the 1:1 stoichiometry observed on three out of four inhibitors tested at this section indicates specificity for a single binding site for the three compounds and therefore assign a priority for further development. We also included explanation of this application of ITC method in the Discussion section.

2. Ebselen was also measured by ITC assay, which was originally reported in the next paragraph (line 115) and now immediately after this. We clarified this point in the revised manuscript accordingly. It is also noteworthy that Ebselen didn't reach saturation until a high molar ratio, which suggested non-specific binding on multiple surface cysteines. K_d was therefore not calculated for ebselen.

119-120: "an IC_{50} " values. Seemingly, some sections were written in non-standard English.

Response: We thank the reviewer for catching this. We have corrected typos or grammar mistakes in the manuscript as mentioned here and below.

P7, 133: Structural formula of GC376 should be shown in a corresponding figure.

Response: Structure of GC376 was indeed shown in Figure 1A. However, we incorporated a new figure demonstrating how GC376 (bisulfite salt) converts to GC373 (aldehyde) and then covalently binds to cysteine of the protease (Supplementary Fig. 7a of the revised manuscript).

P8, 160-161: "as predicted based on chemical logic" is not such a convincing argument. I would prefer can be expected or something similar.

Response: We agree with the reviewer on this. As suggested, we removed this phrase for a more streamlined and concise sentence.

165-169: The sentence is confusing and too long. Thus, It should be split and reworded.

Response: We agree with the reviewer that this sentence might be confusing and too long. Since we no longer report K_d values as derived from ITC data, we have removed this sentence, which was an explanation of reporting K_d for covalent inhibitors. Instead, we discuss our application of ITC method in the Discussion section.

P9, 179: Caspase-3 is only active as dimer, its specificity is Asp in P1. A comparison with the 3CL protease makes no sense and, thus, caspase-3 should be deleted from the article.

182: Also, the specificity of the serine protease chymotrypsin is quite different compared to 3CL, but it is acceptable.

Response: As suggested by the reviewer, we removed the comparison of 3CL protease with Caspase-3, due to the lack of similarity in substrate specificity. Also, as suggested, we kept the chymotrypsin data.

P10,194: It is not clear why EC_{90} values were used, and not EC_{50} . A comparison with the IC_{50} might be easier, otherwise explain it.

Response: We agree with the reviewer that consistently reporting EC50 values throughout the manuscript instead of EC90 values would be more intuitive to readers, especially considering the comparison with IC50 values. We have therefore revised the manuscript to replace all EC90 values by EC50 values whenever possible. The only exception was the additional SARS-CoV-2 antiviral assay performed in Caco-2 cells, as a confirmatory assay. The experiments were conducted by an NIAID-sponsored third-party contractor, who reported EC90 values. However, this is supplemental to the EC50 values measured in the SARS-CoV-2 antiviral assay performed in Huh-7 cells.

Additionally, we used these EC90 values to compare the concentration of the inhibitor in plasma and lungs as measured in a pharmacokinetic study. EC90 values are more relevant in this case because we would aim to achieve a concentration that can block viral replication >90%. We included the explanation in the corresponding text, figure legends, and methods.

It is also noteworthy that, in this particular case, the high antiviral-EC₅₀/biochemical-IC₅₀ ratio observed in Vero cells was an artifact of the high efflux potential of Vero cell line and may underestimate the antiviral potency in human lung cells, the relevant tissue for COVID-19 (J Med Chem 63, 12725-12747, doi:10.1021/acs.jmedchem.0c01063, 2020). Accordingly, we used additional cell lines for the evaluation of antiviral potency during the later lead optimization stage that lack drug efflux.

P13, 247: The abbreviation CYP has to be explained.

Response: We thank the reviewer for reminding us on appropriately using abbreviations. Indeed, for this case, CYP is cytochrome P450, mentioned in line 238 of the original manuscript. As the reviewer suggested, we have checked all abbreviations in the manuscript and made sure we explained each abbreviation upon first usage in the revised manuscript.

265: “vulnerable to metabolism“ sounds a bit weird, do reword it.

Response: As suggested, we have rephrased this to “exhibited lower metabolic stability”.

P14, 270: “didn’t“ is colloquial, it has to be changed to “did not“.

Response: We thank the reviewer for catching this. The sentence has been revised accordingly.

P15, 295: Ki values for covalent inhibitors make no sense. Use kinact/KI (e.g. Krippendorff et al, 2009, J. Biomolecular Screening)

Response: As the reviewer pointed out, inhibition constants (Ki) are not suitable for evaluations of covalent inhibitors, as it works better for non-covalent ligands. Therefore, following the reviewer’s suggestion, instead of Ki, we now use kinact/Ki values to

characterize and compare the most potent covalent inhibitors in the revised manuscript. We also included explanation of this application of k_{inact}/K_i values in the Discussion section.

299: Why now EC50, when before EC90 was used? Consistency is required.

Response: We agree with the reviewer that consistently reporting EC50 values throughout the manuscript instead of EC90 values would work better, as discussed in the response above. We have therefore revised the manuscript to replace all EC90 values by EC50 values whenever applicable.

P16, 316: Now it is EC90 again. Use nM values.

Response:

1. We agree with the reviewer that consistently reporting EC50 values throughout the manuscript instead of EC90 values would work better. We have therefore revised the manuscript to replace all EC90 values by EC50 values whenever applicable. The only exception is this particular case, which is the additional SARS-CoV-2 antiviral assay performed in Caco-2 cells, as a confirmatory assay. The experiments were conducted by an NIAID-sponsored third-party contractor, who only reported EC90 values as results. However, this could be supplemental to the EC50 values measured in SARS-CoV-2 antiviral assay performed in Huh-7 cells. Additionally, we later used these EC90 values to compare the concentration of the inhibitor in plasma and lungs as measured in the pharmacokinetic study. EC90 values are more relevant in this comparison because we would aim to achieve a concentration that can block viral replication to >90% efficacy. We included the explanation in the corresponding text, figure legends, and methods.

2. As suggested, we have used nM instead of μM as units in the revised manuscript.

P17, 327: Not “1 equivalent“, but one equivalent; analogs.

Response:

We thank the reviewer for catching this. The sentence has been revised accordingly.

P17, 331: I wonder, whether it is possible to obtain a true K_d for covalent inhibitors from ITC. I doubt it seriously and think it must be a counterpart to k_{inact}/K_i .

Response:

We agree with the reviewer that disassociation constants (K_d) are not suitable for evaluations of covalent inhibitors, as it can only provide the apparent binding affinities for covalent ligands and works better for non-covalent ligands. Instead, following the reviewer's suggestion, we now use k_{inact}/K_i to characterize and compare our most potent covalent inhibitors in the revised manuscript. For ITC data, we now only report stoichiometry and binding energy, but not “ K_d ”, in the revised manuscript. We also included explanation of this application of ITC method in the Discussion section.

P18, 357/367: did not

Response:

We thank the reviewer for catching this. The sentence has been revised accordingly.

P19, 378-381: The sentence is confusing, rewrite it.

Response: As suggested, the sentence has been revised accordingly.

391: “Strong no-polar“? How is the strength measured? Is it just a hydrophobic interaction, which is usually not that strong.

Response: As suggested, we have rephrased this sentence to clarify it is a hydrophobic interaction as observed and deduced from the co-crystal structure.

P20, 394-396: Sentence requires rewording.

Response: As suggested, we did break this long sentence into two shorter sentences for clarity.

398-399: “covalently but reversibly“ seems contradictory, while aldehydes may be over time hydrolyzed. Anyway, some proof is needed, not simple conjectures.

Response: It has been demonstrated in the literature that the lead compound GC376 converts to an aldehyde and then covalently forms a thioester with the active site cysteine (<https://www.nature.com/articles/s41467-020-18096-2>). Such a thioester bond is sensitive to solution conditions (pH etc.) and may therefore be hydrolyzed. Our mass spectrometry study of the binding of GC376 also demonstrated this point. However, “covalently but reversibly” does appear to be confusing to readers. So as suggested, we have removed it and made the text more concise. We thank the reviewer.

P21-22, 435-439: The long sentence is not very clear.

Response: As suggested, we did break this sentence into three shorter sentences for clarity.

P22, 444: Why is not EC50, use nM values.

Response: We agree with the reviewer that consistently reporting EC50 values throughout the manuscript instead of EC90 values would work better. As suggested, we have therefore revised the manuscript to replace all EC90 values by EC50 values whenever applicable, including this case. Also, nM values are now used instead of μM values.

448: within the cellular context.
451: coronaviruses

Response: We thank the reviewer for catching these points. We have revised the text accordingly. We really appreciate the detailed reviews and suggestions provided by the reviewer.

P23, 464-465: EC90 again, should 100 nM; with an EC50 value.

Response: We have revised the text as the reviewer suggested (“100 nM”; “with an EC50 value”).

476: IND is not explained.

Response: We thank the reviewer for reminding us to avoid using abbreviations without explanation upon first usage. We have checked all abbreviations in the manuscript and made sure we explained each of the abbreviations upon first usage in the revised usage. In this particular case, we explained IND as Investigational New Drug.

P24-25: The Discussion is a mere summary with an outlook. This section requires extensive rewriting.

Response: We agree that the discussion section needed to be adjusted. Thus, we have fully revised the discussion section, including the discussion of the methods and materials being used.

P43, 872/878: Kd for covalent inhibitors appears to be inappropriate.
874/876: I guess “heat” is not the proper expression, so I assume it is the free energy.

Response:

1. As the reviewer pointed in the comments, disassociation constants (Kd) are not ideal for evaluations of covalent inhibitors, as it can only provide the apparent binding affinities for covalent ligands and works better for non-covalent ligands. Instead, following the reviewer’s suggestion, we used kinact/Ki to characterize and compare our most potent covalent inhibitors in the revised manuscript. For ITC data, we now only reported stoichiometry and binding energy, but not “Kd”, in the revised manuscript. We also included explanation of this application of ITC method in the Discussion section.
2. Heat was technically measured by ITC to calculate free energy.

P44, 895: Standard for IC50 data is fitting log(inhibitor) vs normalized response. Asymmetric sigmoidal curves might completely distort the true values. I suggest to recalculate all IC50 data accordingly.

Response: We really appreciate the reviewer for catching this, as our IC50 data were indeed fitted Log(inhibitor) vs. normalized response curve, the standard inhibition curve. But we incorrectly described this in the Method section. We did correct and clarify our description of the methods in the Method section. And as the reviewer suggested, we checked each IC50 curve for the accuracy.

906: The standard for enzyme kinetics is triplicate data, so repeat the experiment accordingly.

910: Ki values for covalent inhibitors cannot be calculated properly, use kinact/KI values.

Response:

1. As the reviewer pointed out, inhibition constants (Ki) are not suitable for evaluations of covalent inhibitors, as it works better for non-covalent ligands. Instead, following the reviewer's suggestion, instead of Ki, we now use kinact/KI values to characterize and compare our most potent covalent inhibitors in the revised manuscript. We also included explanation of this application of kinact/KI values in the Discussion section.

2. As the reviewer pointed out, the standard for enzyme kinetics uses triplicate data. So the measurement of kinact/KI values were conducted in triplicates accordingly.

P62, 1314: Nowadays, it is standard to deposit structural data in the Protein Data Bank (PDB) upon submission of the manuscript. Thus, the final PDB codes have to be included in the manuscript. Of course, the corresponding PDB data should have the status "hold for publication" and are only released, when the paper is accepted/published.

Response: We thank the reviewer for kindly reminding us about this. We have now deposited all protein crystal structural data report in this manuscript into Protein Data Bank. The accession codes PDB IDs are 7TIU, 7TIV, 7TIW, 7TIX, 7TJ0, 7TIA, 7TIY, and 7TIZ. The corresponding PDB data are currently hold for publication and will be released upon publication of this manuscript.

P66-69: The figures 1, 2, 3 and 5 are far too small, which applies to the molecule structures and especially to all the tiny labels. New, larger versions have to be prepared.

Response: As suggested, we have adjusted the appearance of these figures for better visualization in the revised manuscript.

P68, 1394: I am not familiar with the standard procedures of ITC, but duplicates seem dubious to me. I would prefer triplicates.

Response: We appreciate the understanding of reviewer on performing duplicate experiments on ITC (as mentioned in the points below). This is based on no significant

variations observed from the duplicates. Additionally, binding analysis by ITC is primarily for binding stoichiometry analysis (1:1 binding). And ITC data is supplementary to the determination of k_{act}/K_I values in kinetic experiments for covalent inhibitors.

1402-1407: eq should be written out at least for the first occurrence.
P69, 1423: What is MOI?

Response: We will explain all abbreviations in the manuscript upon first usage
Response: We thank the reviewer for reminding us to avoid using abbreviations without explanation upon first usage. We have checked all abbreviations in the manuscript and made sure we explained each of the abbreviations upon first usage in the revised usage. In this particular case, eq is explained as equivalence and MOI stands for “multiplicity of infection”.

1427: mouse – which cell type?

Response: CD-1 mouse plasma and liver microsome were used in this experiment. We have included this information in the figure legends and Method section of revised manuscript.

P70: Table 1. Display IC₅₀ values, not K_i. The resolution of the crystal structure does not contribute any information on inhibition. Delete the column.

Response:

1. As suggested, we now display IC₅₀ values instead of K_i values.
2. As suggested, we have deleted the column with the resolution of the crystal structures.

P71: The labels for the residues have to be enlarged.

Response: As suggested, we have adjusted the appearance of these figures for better visualization.

P74-75: Extended data Figures 1 and 2. As in previous figures the labels are too small. The molecular structures are nearly invisible, data points and curve quite faint. Why are they shown as dose response curves? Show the plots as is standard. I guess that sigmoidal plots were applied not standard log (conc inhibitor) ones. Plots 2, 4, 8, 17, 19, 20, 21, 31, 32, 36, 38, 39, 42, 43, 44, 49, 52, 53, 54, and 56 make not much sense. Either replot them as log curves, or remeasure them. Otherwise delete them. It is impossible to calculate proper IC₅₀ with this approach, when so many curves start at 25% or even 50% inhibition.

Response: We agree with the reviewer that the original Extended data Figures 1 and 2 need to be better organized, especially for the molecular structures and the fitting of dose-response curve. As suggested, we now significantly enlarged all molecular structures. Instead of showing dose-response curve, we show the inhibition effects at 10

μM and $100 \mu\text{M}$ concentrations of each compound. This is based on our observation and definition that a validated screening hit should be able to completely inhibit 3CL protease activity at $100 \mu\text{M}$. Furthermore, a top hit (potent inhibitor) should be able to inhibit $>50\%$ of the 3CL protease activity at $10 \mu\text{M}$ (if $\text{IC}_{50} < 10 \mu\text{M}$). GC376 is included as a positive control in this dataset. Qualified potent inhibitors are highlighted in green. Detailed data at more concentrations with more information are reported in source data for Supplementary Figure 1 and 2.

P76-78: Extended data Figures 3, 4, and 5. Labels are too small. Figure 5D: A comparison with chymotrypsin might be acceptable, because of the fold. However, elastase would be the better choice as the specificity is more similar. Caspase-3 is a bad choice and should be replaced by Cathepsin B or preferentially Cathepsin L.

Response:

1. The labels in Extended data Figures 3, 4, and 5 are enlarged for better visualization.
2. We agree with the reviewer on the selection of human protease for off-target effect check. Accordingly, we keep the comparison with chymotrypsin but removed Caspase-3.
3. For elastase, Cathepsin B, and Cathepsin L, we tested our top compounds against these proteases (as reported in Supplementary Table 1 of the revised manuscript).

P80: Extended data Figure 6. In panels A, B and C the labels are just ok, while in D, E and F the are too small again. In 6D K_i values are displayed, which is basically not possible for covalent inhibitors, calculate k_{inact}/K_i .

Response:

1. As suggested, the appearance all panels were adjusted for better visualization.
2. We agree with the reviewer that K_i values are not suitable for describing covalent inhibitors. Therefore, we have replaced K_i values with IC_{50} values.

P81, 1602: Duplicates are problematic, perhaps acceptable for ITC.

Response: We appreciate the understanding of reviewer on performing duplicate experiments on ITC. This is based on no significant variations were observed from the duplicates. Additionally, binding analysis by ITC is primarily for binding stoichiometry analysis (1:1 binding). And ITC data is supplementary to the determination of IC_{50} values in biochemical activity assays.

P82: Extended data Figure 7. All panels require general enlargement.

P84: Extended data Figure 8. Panel A has to be enlarged, while B and C are ok.

Response: As suggested, all panels of Extended data Figure 7 and 8 have been enlarged in the revised manuscript for better visualization.

P87-88: Extended data Figure 9: Except for panels B and C significant enlargement is

required. 1723-1724: Ki problem and only duplicates; 1743: The duplicate in ITC might be ok.

Response:

1. As suggested, we have enlarged the corresponding panels in Extended data Figure 9 for better visualization.

2. We agree with the reviewer that Ki values are not suitable for describing covalent inhibitors. Therefore, we have replaced Ki values with kinact/KI values. The kinetic experiments were performed in triplicates

3. We appreciate the understanding of reviewer on performing duplicate experiments on ITC. This is based on no significant variations were observed from the duplicates. Additionally, binding analysis by ITC is primarily for binding stoichiometry analysis (1:1 binding). And in our study, ITC is supplementary to the determination of kinact/KI values in kinetic experiments for covalent inhibitors.

P90: Extended data Figure 11. 2Fo-Fc and Fo-Fc maps cannot be distinguished for all panels. The 3s contour is unusual for the 2Fo-Fc map (typically 1s). The figures would be more convincing, if 2Fo-Fc omit maps were shown, calculated without the inhibitor model.

Response: Our apologies for our unintended mistake in the figure caption, and thanks very much for noting this. Indeed, the electron density mesh (for all panels) is depicting Fo-Fc omit map contoured at 3 sigma. We have revised the figure legends to clarify this.

P92-93, 1816-1817: Extended data Figure 12. Correct to “and the respective 3CL protease in green”

Response: We thank the reviewer for catching this. We have edited the text as the reviewer suggested.

P94-95: Extended data Figure 13. Labels in panels B and D are just a bit to small, the rest is ok. 1852-1854: The Ki problem and only duplicates.

Response:

1. As suggested, we have enlarged panels B and D for better visualization.

2. We agree with the reviewer that Ki values are not suitable for describing covalent inhibitors. Therefore, we have replaced Ki values with IC₅₀ values.

P96: Extended data Table 1. It is alright to analyze for all these proteases with respect to interference of the compounds. Obviously, Cathepsins B and especially L are affected by the compounds EB54 and NK01-63. They are cysteine proteases and their

specificity is closer to the 3CL protease than chymotrypsin or caspase-3 according to the MEROPS database. 1867: Better show the IC50 value for the 3CL protease.

Response: As suggested, we now show the IC50 values for 3CL protease.

P98: Extended data Table 3. Z-FR-AMC is certainly not the best substrate for Cathepsin B and L. In case of CatB Gly and Ala are preferred in P1 position, CatL prefers Gly, Ala, Ser, and Thr.

Response: As the reviewer suggested, Z-FR-AMC might not be the best substrate for Cathepsin B and L, considering the protease preference in P1 position of substrate. Here, we used Z-FR-AMC in the enzyme assay because it is a commercially available substrate and has been commonly used in Cathepsin L assays in the literature (Sacco et al, Science Advance, 2020, DOI: 10.1126/sciadv.abe0751; Liu et al, Antiviral Research, 2021, DOI: 10.1016/j.antiviral.2021.105020).

However, following the reviewer's suggestion, we tested two commercially available fluorogenic peptides for whether they can be used as Cathepsin L substrates: Z-RLRGG-AMC and MCA-PLGL-Dap(Dnp)-AR-NH2. Z-RLRGG-AMC can specifically detect cleavage with Gly in P1 position, while MCA-PLGL-Dap(Dnp)-AR-NH2 is known to be cleaved with Gly in P1 position when used in other enzyme assays.

The results showed that Cathepsin L is not active on the Z-RLRGG-AMC substrate, but is active on the MCA-PLGL-Dap(Dnp)-AR-NH2 substrate. Cathepsin L was less active with MCA-PLGL-Dap(Dnp)-AR-NH2 when compared to Z-FR-AMC. Therefore, when performing the IC50 assays, 0.5 nM Cathepsin L was used with MCA-PLGL-Dap(Dnp)-AR-NH2 substrate, while 0.042 nM Cathepsin L was used with Z-FR-AMC substrate. As a result of the higher concentration of Cathepsin L being used in the assay, IC50 values of EB54 and NK01-63 increased (as shown in the extended data table 1). These commercially available substrates might not be the best substrates due to their sequences, but they consistently support the observation reported in the manuscript that: the tested compounds also have inhibitory effects on Cathepsins, which might block SARS-CoV-2 infection and replication via dual inhibition of 3CL protease and cathepsin L, and may thus act as multi-targeted antivirals.

Reviewer #2 (Remarks to the Author):

Dear authors:

Liu et al designed a series of SARS-CoV-2 inhibitors based on the previous coronavirus 3CL protease inhibitor GC376. Binding assay, cellular antiviral assay and co-crystallization all showed they have promising results. Furthermore, these compounds have pan-coronavirus activity. Among them, NK01-63 is promising. Overall, this study is technically sound, the results are interesting in the field of the drug

discovery for COVID-19, and the manuscript well written. Development of an effect antiviral is very important to control the pandemic. The limitation is lack of *in vivo* study, including the stability, toxicity and metabolism of the compounds in animal, efficacy of treatment on infected animals.

Response: We thank the reviewer for the positive comments. We strongly agree with the reviewer that *in vivo* study will further enhance our manuscript. Therefore, we evaluated our top compound NK01-63 *in vivo* for toxicity and pharmacokinetic study, and included data in the revised manuscript.

Specific questions:

1. Since they are free-cystein reactive, I suggest authors to test the inhibition against other important human cystein proteases such as caspase-3, cathepsin K, S etc.

Response: We agree with the reviewer that we should test the inhibition of these compounds against important human cysteine proteases. We have done so against human cysteine protease Caspase-3, Caspase-8, Cathepsin B, and Cathepsin L. The inhibition data were reported in the original manuscript (Extended Data Table 1). Furthermore, following the suggestion of the reviewer, we further tested the inhibition against cathepsin K and S. The results were summarized in the revised manuscript (Supplementary Table 1). In summary, besides SARS-CoV-2 3CL protease, the most outstanding inhibition by NK01-63 was observed on human Cathepsin L. However, since studies have indicated that cathepsin L inhibitors can substantially decrease SARS-CoV-2 viral entry without showing toxicity to the host, NK01-63 is expected to effectively block SARS-CoV-2 infection and replication via dual inhibition of 3CL protease and cathepsin L, and may thus act as multi-targeted antivirals. Second, the low IC₅₀ value observed on Cathepsin L might be a result of the low concentration of Cathepsin L used in the assay (0.042nM). Increase of the concentration of Cathepsin L used in the assay to 0.5 nM significantly increased the observed IC₅₀ values. In addition, the absence of cytotoxicity as observed in cells *in vitro* and no toxicity shown *in vivo* also supported the clinical usage of NK01-63 for antiviral therapeutics.

2. *In vivo* toxicity is still of my concern. If possible, I suggest authors at lease evaluate the short term toxicity in animals.

Response: We agree with the reviewer that *in vivo* toxicity study will further enhance our manuscript. Therefore, we evaluated our top compound NK01-63 *in vivo* for toxicity, and included data in the revised manuscript. In particular, we monitored the body weight change of C57BL/6 mouse treated with 20 mg/kg NK01-63 or water vehicle via intraperitoneal (IP) or oral (PO) dose for 14 consecutive days. No significant change in body weight was observed as compared to the vehicle group, showing that NK01-63 has no *in vivo* toxicity via either route of administration. In addition, we also performed a pharmacokinetic study of NK01-63, in which we find half-life of NK01-63 in critical tissues such as lung is long, so that the concentration of NK01-63 in lung after 24 hours of treatment is still above its cellular EC₉₀ value against SARS-CoV-2. We think these *in vivo* data further support the clinical potency of GC376 analogs, such as NK01-63.

3. I'd suggest authors to compare with other first line antivirals such as Remdesivir in vivo and in vitro.

Response: We agree with the reviewer that comparison with other first line antivirals will enhance our manuscript. Therefore, we compared NK01-63 with currently developed SARS-CoV-2 3CL protease inhibitors and summarized the comparison in Extended Data Table 2 of the originally submitted manuscript. As suggested by the reviewer, we further updated the table to include recent progress on SARS-CoV-2 3CL protease inhibitors. Among these compounds, two SARS-CoV-2 3CL protease inhibitors developed by Pfizer Inc., PF-00835231 and PF-07321332, appeared to be the most advanced in clinical applications. PF-00835231 has entered clinical trial in 2020, while the combination of PF-07321332 (also known as Nirmatrelvir) with ritonavir has recently been granted emergency use authorization by the US FDA. However, the side-by-side comparison showed that NK01-63 (*Coronastat*) features the most outstanding potency among all reported SARS-CoV-2 3CL protease inhibitors currently in development. We therefore propose performing further preclinical studies on NK01-63 (*Coronastat*) to evaluate its potential for clinical development.

On the hand, the first antiviral Remdesivir is a RNA-dependent RNA polymerase inhibitor, which has a different mechanism of action compared to our inhibitors. Although direct comparison of Remdesivir with 3CL protease inhibitor might be less applicable, recent study indicated that 3CL protease inhibitor (such as PF-00835231) has additive/synergistic activity in combination with Remdesivir (Boras et al, Nature Communications, 2021, doi: <https://doi.org/10.1038/s41467-021-26239-2>). We therefore envision that combination of our inhibitors with other first line antivirals to be evaluated for clinical development.

As suggested by the reviewer, we have included this comparison in the discussion section of the revised manuscript. We appreciate the suggestions provided by the reviewer to further enhance our manuscript.

Reviewer #3 (Remarks to the Author):

This manuscript describes extensive studies on analogs of GC376 and other small molecules as inhibitors of the main protease (3CL) of the SARS2 virus that causes COVID19. The work is well done, and the authors have found several potent inhibitors that show promising properties as drugs, including the compound they call coronastat.

Response: We sincerely thank the reviewer for the positive comments.

The main difficulty with this manuscript is lack of novelty.

After the initial publication of Hilgenfeld and those of other groups in 2020 (their refs 1-10 in the submitted paper), a number of analogs of GC376 were examined for enzyme inhibition, by crystallography of enzyme-inhibitor complexes, and for antiviral activity in cells (e.g. ref 26 in their paper). Some of these analogs, including from ref 26, are re-reported without citation in the present paper as new experimental procedures and results. This is not acceptable. This reviewer doubts that the incremental advances in the present paper merit publication in this journal. The authors use the word "we" very extensively in their paper, often disguising the fact that much has been done before and they are simply repeating previous publications.

Response: We strongly agree with the reviewer that we should appropriately cite all recent works on GC376 analogs and reasonably compare these compounds side by side. Indeed, we did cite most recent works with exciting results on 3CL inhibitors and we did prepare a table to summarize all top compounds for direct comparisons, as in Extended Data Table 2 of our original manuscript in the initial submission. This table did include top compound 2c and 2d from ref.26 (Vuong, W. et al. European Journal of Medicinal Chemistry 2021, doi: <https://doi.org/10.1016/j.ejmech.2021.113584>).

We appreciate the contributions of ref.26 to the field of COVID research, we note that our preparation and characterization of EB34 (known as compound 1e in ref.26), NK01-14 (as compound 2a in ref.26), and NK01-57 (as compound 2d in ref.26) was completely independent and was completed long before the publication of ref.26, as documented by our lab notebooks and journal submission records. We view ref.26 as evidence supporting the effectiveness of our optimization strategy of GC376. NK01-63 showed further improvement over NK01-57 (best compound 2d in ref.26) in the side-by-side comparison in the kinact/KI enzymatic kinetic assays and also cellular anti-viral experiments. While we think this demonstrates the strength of our manuscript, we envision that our research efforts, jointly with ref.26 as well as other work in the field, will eventually lead to globally available and effective small molecule anti-viral treatments for COVID-19. We appreciate that the reviewer points out this citation, and we have clarified this point in the revised manuscript and further incorporated a discussion to summarize recent work on GC376 for future studies.

There is another problem. Their claim that "...GC376 was soluble in PBS buffer at all concentrations tested (up to 1 mM, Fig. 3c.." is not agreement with other studies. That is why it is injected in aqueous ethanol solution into cats and other animals in the literature. It does form clear micelles and colloid suspensions when concentrated in water (see ref 26 in their paper) but such administration to humans requires additional testing in animals. Injection of aqueous ethanol solutions or DMSO solutions into humans is not feasible.

Response: We need to clarify that in our manuscript the solubility of GC376 was tested in PBS with 1% DMSO (v/v) on a nephelometer via monitoring turbidity (Figure 3C in the original manuscript). This is different from the solubility test method in ref.26, in which

pure water was used as solvent and visual observation was used. We reasoned that the difference in solvents, especially the presence of 1% DMSO, may be the origin of the different result, as we observed no significant turbidity for GC376 samples when compared to Compound 4 (Figure 3C). But we do appreciate the reviewer pointed out that we didn't clearly demonstrate the solvent effect in the text. We have revised our text for clarity. Also, as suggested, we further tested the solubility of GC376 in pure water at higher concentrations (1 mM - 800 mM). Although turbidity was observed in water solutions with 25mM - 200mM GC376, the solution became clear at 400 mM and 800 mM (Extended Data Fig. 12 in the revised manuscript). This is consistent with the observation in ref.26, and is potentially due to micelle formation as reported in ref.26. As ref.26 pointed out in their interpretation, such micelle solutions may be a possible method for circumventing solubility issues and volume limits when considering GC376 for use *in vivo*. Therefore, for the formulation of our best compound, NK01-63, for *in vivo* studies, we decided to use pure water as solvent. The concentration of NK01-63 treatment was 2 mg/ml (3.5 mM) in water for both IP and PO dosage in C57BL/6 mouse. We found the solution was clear and no toxicity was observed after either route of administration. A pharmacokinetic study demonstrated delivery of the compound in this formula to plasma and critical tissues such as lung at concentrations of $>100 \times EC_{90}$ values. Thus, we consider formulation with water to be suitable. As suggested by the reviewer, we have revised the text to include this discussion.

Reviewers' Comments:

Reviewer #1:

Remarks to the Author:

After thoroughly reading the response letter of Liu and his coworkers and the relevant parts of the revised version, I see that they have done a meticulous job to rework all critical points of my first review. Seemingly, in this line they did it for the critical comments of the other reviewers as well. As a structural biologist with a background in enzyme kinetics, I mostly appreciate that submitted the structures to the PDB and received the used k_{inact}/K_i values systematically. These two examples may suffice, without repeating every point of the response letter. Therefore, I will now suggest that the greatly improved manuscript is accepted for publication in Nature Communications.

Reviewer #2:

Remarks to the Author:

Dear authors:

Thanks for your revision. This version has addressed most of my concerns. Therefore, I'd like to recommend acceptance of your manuscript for publication.

Revision of manuscript NCOMMS-21-31169A

The following document includes **a point-by-point response to the referee concerns** and our corresponding revisions.

REVIEWERS' COMMENTS

Reviewer #1 (Remarks to the Author):

After thoroughly reading the response letter of Liu and his coworkers and the relevant parts of the revised version, I see that they have done a meticulous job to rework all critical points of my first review. Seemingly, in this line they did it for the critical comments of the other reviewers as well. As a structural biologist with a background in enzyme kinetics, I mostly appreciate that submitted the structures to the PDB and received the used k_{inact}/K_i values systematically. These two examples may suffice, without repeating every point of the response letter. Therefore, I will now suggest that the greatly improved manuscript is accepted for publication in Nature Communications.

Response: We sincerely appreciate the constructive and positive comments provided by the reviewer during the revision.

Reviewer #2 (Remarks to the Author):

Dear authors:

Thanks for your revision. This version has addressed most of my concerns. Therefore, I'd like to recommend acceptance of your manuscript for publication.

Response: We sincerely appreciate the constructive and positive comments provided by the reviewer during the revision.